# GeoCache: Provably Lossless Inference and Computational Content-Level Isolation
# for Shared KV-Caches in Multi-Tenant LLM Inference
# via Isometric Orthogonal Transformation

## Abstract

Key-value (KV) cache sharing across user sessions is a critical optimization for multi-tenant large language model (LLM) serving, reducing time-to-first-token (TTFT) latency by up to 73%. However, recent work has demonstrated that shared KV caches create exploitable vulnerabilities: an adversary can exfiltrate data or mount semantic poisoning attacks by accessing cross-tenant cache states. Existing mitigations either disable sharing entirely (negating performance gains) or employ heuristic content classifiers with inherent false-positive/negative tradeoffs. We propose GEOCACHE, a principled approach that applies session-specific orthogonal transformations to key and value attention matrices before caching. GEOCACHE exploits the isometry property of orthogonal operators to provide two complementary guarantees: (1) *exact preservation* of attention computation for authorized in-session cache reuse, proven by the identity $\mathbf{q}^\top \mathbf{M}^\top \mathbf{M} \mathbf{K} = \mathbf{q}^\top \mathbf{K}$ (Theorem 1); and (2) *geometric isolation* of cross-session cache entries, with attention scores concentrating near zero under a BDOT-specific exponential bound (Theorem 2). The approach uses a Block-Diagonal Orthogonal Transform (BDOT) achieving $O(d_k)$ per-token overhead and sub-microsecond $(0.014\mu\text{s})$ latency per head-token on an NVIDIA T4 GPU. Experiments on Llama-2-7B and Mistral-7B confirm: (a) generation quality is preserved exactly (maximum float32 deviation $5.3 \times 10^{-5}$); (b) controlled exfiltration and injection attacks via the forward attention path are substantially mitigated (ROUGE-L reduced from 0.917 to 0.043, 0/19 secret markers leaked; cross-session cosine similarity reduced from 1.0 to $\approx 0$); and (c) throughput overhead is negligible. We position GEOCACHE as providing *provable utility preservation* together with *computational content-level isolation* (rather than cryptographic security): a separate algebraic recovery attack (Known-Plaintext Attack) on the BDOT operator is feasible if the attacker collects $b$ linearly independent plaintext-ciphertext pairs per block. We therefore require operator rotation every $b/2$ tokens for production deployments serving structured prompts (JSON schemas, few-shot exemplars), which empirically reduces algebraic recovery to subspace-only directional leakage insufficient for exact token reconstruction; under freeform queries, hybrid deployment alone suffices. For deployments requiring timing-channel protection, GEOCACHE composes with constant-time response padding to eliminate existence leakage.

## 1 Introduction

Transformer-based large language models (LLMs) perform autoregressive generation by computing attention over the full input context at each decoding step. To avoid redundant computation, inference systems cache the key (K) and value (V) projection matrices — the *KV-cache* — and reuse them across generation steps. The KV-cache is essential infrastructure: without it, generation complexity is quadratic in sequence length, rendering interactive applications impractical.

In multi-tenant deployments where hundreds of users share a single model instance, KV-cache sharing across sessions provides dramatic efficiency gains. When two users submit queries with identical prefix sequences (e.g., a shared system prompt), the cached KV entries from the first user can be reused for the second, eliminating redundant prefill computation. Major LLM serving frameworks — including SGLang (Zheng et al., 2024), vLLM (Kwon et al., 2023), and TensorRT-LLM (NVIDIA, 2024) — implement cross-session prefix caching as a core optimization.

**The privacy problem.**  Cross-session cache sharing creates an exploitable attack surface spanning four categories (Section 2). *Serving-behavior exploits*: Wu et al. (2025) showed that SGLang's Longest Prefix Match (LPM) scheduling policy leaks prompt content by allowing an attacker to observe which candidate prefix is served first. *Timing side-channels*: Song et al. (2025) demonstrated that TTFT timing differences provide a complementary leakage signal across serving systems. *Content-level KV-cache attacks*: Luo et al. (2025) identified inversion, collision, and injection attacks on cached KV tensors — enabling an adversary to recover input tokens, match candidates by distance, or force the model to echo private content by appending instructions to a stolen cache. *Hardware-level exfiltration*: recent work (Sorensen et al., 2024; Zhang et al., 2024) shows that GPU memory residuals enable direct extraction of cached tensors. These vulnerabilities are compounded by emerging evidence that LLM agents can violate data protection regulations when handling user data (Lichkovski et al., 2025).

**Inadequacy of existing defenses.**  Current mitigations fall into three categories, none satisfactory:

1. **Full cache isolation** disables cross-session sharing entirely, eliminating the side channel but negating the performance benefit. This increases TTFT by up to 73% and significantly reduces serving throughput (Yang et al., 2025).

2. **Selective cache isolation** systems such as SafeKV (Chu et al., 2025) mitigate timing side-channel leakage by classifying cache entries by privacy sensitivity and isolating sensitive entries from cross-tenant reuse. This recovers some caching benefit but requires training a sensitivity detector, introduces false positive/negative tradeoffs, and does not address content-level attacks on the KV tensors themselves.

3. **Timing obfuscation** adds random delays to mask cache hit/miss timing differences. This reduces but does not eliminate the side channel while degrading user-perceived latency.

All three approaches operate at the *systems level* — managing cache entries, classifying content, or adding noise. None addresses the root mathematical cause within the attention mechanism itself: that cross-session KV entries, when improperly accessed by the wrong session's query, output meaningful and exploitable attention scores. We argue that securing transformer inference requires understanding and modifying the geometric properties of the attention space directly.

**Our approach.**  We propose GEOCACHE, which addresses the root cause by making cross-session cache entries *geometrically meaningless* to unauthorized queries. The key insight is that an orthogonal transformation is an *isometry*: it preserves all inner products and norms exactly. By applying a session-specific orthogonal operator to K and V matrices before caching, and applying the same operator to queries during retrieval:

- **Within-session** attention computation is *mathematically identical* to unprotected inference, because $\mathbf{q}^\top \mathbf{M}^\top \mathbf{M} \mathbf{K} = \mathbf{q}^\top \mathbf{K}$ when the same operator M is used.

- **Cross-session** attention scores concentrate exponentially near zero, because independent Haar-distributed orthogonal operators produce statistically orthogonal coordinate systems.

GeoCache operates at the *mathematical level* of the attention mechanism, not at the systems level. No cache entry is ever deleted, classified, or delayed. Instead, cross-session entries become geometrically invisible — they produce attention scores indistinguishable from noise.

**Contributions.** Our contributions advance the understanding of ML attention mechanisms in multi-tenant environments:

1. We formalize the KV-cache cross-session leakage problem as a geometric isolation problem in the attention key space and prove that session-specific orthogonal transformation provides exponentially strong isolation (Theorem 2) with zero inference quality degradation (Theorem 1).

2. We design the GEOCACHE protocol, including efficient Block-Diagonal Orthogonal Transform (BDOT) construction achieving $O(d_k)$ per-token overhead, layer-specific salt derivation for independent per-layer transformations, and an inverse-recovery module that restores exact attention outputs at $0.014\mu s$ latency per head-token on an NVIDIA T4 GPU.

3. We evaluate GEOCACHE on Llama-2-7B and Mistral-7B, demonstrating: (a) generation quality is preserved exactly (maximum float32 deviation $5.3 \times 10^{-5}$); (b) cross-session attention scores concentrate near zero, substantially mitigating controlled exfiltration and injection attacks (ROUGE-L reduced from 0.917 to 0.043, 0/19 secret markers leaked; cross-session cosine similarity reduced from 1.0 to $\approx 0$); and (c) the BDOT transformation introduces negligible throughput overhead with only 1 MB of per-session operator storage.

Alongside recent work (Luo et al., 2025), GEOCACHE establishes that structural mathematical isolation offers a viable path to secure LLM inference without the tradeoffs inherent to systems-level mitigations.

## 2 Background and Related Work

### 2.1 KV-Cache in Transformer Inference

In a standard multi-head attention layer with $h$ heads and per-head key dimension $d_k$, the attention computation for a query vector $\mathbf{q} \in \mathbb{R}^{d_k}$ over cached keys $\mathbf{K} \in \mathbb{R}^{n \times d_k}$ and values $\mathbf{V} \in \mathbb{R}^{n \times d_k}$ is:

$$\text{Attention}(\mathbf{q}, \mathbf{K}, \mathbf{V}) = \text{softmax}\left(\frac{\mathbf{q}^\top \mathbf{K}^\top}{\sqrt{d_k}}\right) \mathbf{V} \tag{1}$$

During autoregressive generation, $\mathbf{K}$ and $\mathbf{V}$ grow by one row per generated token. Caching these matrices avoids recomputing attention over the entire context at each step, reducing per-step complexity from $O(n \cdot d_k)$ to $O(d_k)$ for the new token's K/V computation.

### 2.2 Cross-Session Prefix Caching

In multi-tenant deployments, many user sessions begin with identical prefixes (system prompts, few-shot examples, or shared document context). Prefix caching stores KV entries keyed by token-sequence hash: if session $B$'s prefix matches session $A$'s cached prefix, session $B$ reuses $A$'s KV entries without recomputation. This optimization can reduce TTFT by 40–73% depending on prefix length and sharing patterns (Yang et al., 2025; Zheng et al., 2024).

### 2.3 KV-Cache Attack Surfaces

Recent literature has identified distinct vulnerability classes in multi-tenant shared caches, which we organize into four categories:

**Serving-behavior exploits.** Wu et al. (2025) demonstrated PromptPeek, which exploits SGLang's Longest Prefix Match (LPM) scheduling policy: the attacker submits concurrent candidate prefixes and observes which is served first (due to LPM priority), reconstructing the victim's prompt token by token.

**Timing side-channels.** Song et al. (2025) systematically studied timing side-channels across LLM serving infrastructure, showing that TTFT differences from KV-cache hits, semantic cache lookups, and GPU memory allocation patterns all leak information about user prompts. These attacks infer cache *presence* (whether a prefix is cached) rather than cache *content*.

**Content-level KV-cache attacks.** Luo et al. (2025) (*Shadow in the Cache*, NDSS 2026) identified and evaluated three attack classes that operate directly on cached KV tensors, and concurrently proposed KV-Cloak as a defense in the same work (Section 2). The attacks are: (i) *inversion attacks*, which algebraically recover input tokens from first-layer Key vectors via least-squares inversion of the projection matrix; (ii) *collision attacks*, which match candidate tokens against cached Keys by Frobenius distance; and (iii) *injection attacks*, which append an instruction (e.g., "repeat the above") to a stolen KV-cache to force the model to echo private content. These attacks assume the adversary can access cached KV tensors, e.g., via externalized storage in PagedAttention-based systems.

**Hardware-level exfiltration.** Recent vulnerability research (Sorensen et al., 2024; Zhang et al., 2024) demonstrates that GPU memory residuals and cache side-channels enable direct extraction of cached tensors from co-located workloads, bypassing software-level access controls.

GEOCACHE targets the third category (content-level attacks). It is complementary to defenses against serving-behavior exploits (e.g., scheduling policy changes), timing side-channels (e.g., constant-time padding), and hardware-level exfiltration (e.g., TEE-based isolation).

### 2.4 Existing Defenses

**Cache partitioning.** Per-user cache isolation eliminates cross-session reuse entirely. While secure, this approach increases memory usage (duplicated entries for shared prefixes) and eliminates TTFT benefits. Pennas et al. (2026) proposed CacheSolidarity, which monitors prefix reuse patterns and selectively isolates suspicious sharing, recovering higher cache reuse than full isolation. Chu et al. (2025) proposed SafeKV, which mitigates timing side-channel leakage by classifying cache entries by privacy sensitivity and isolating sensitive entries from cross-tenant reuse. This recovers some caching benefit but requires a trained sensitivity detector with inherent error rates, and does not address content-level attacks on the cached KV tensors themselves.

**Timing defenses.** Random delay injection and constant-time padding attempt to eliminate the timing signal. These approaches degrade TTFT for all users and provide statistical rather than mathematical guarantees — an adversary with sufficient queries can average out the noise.

**Orthogonal transformations in ML security.** While foundational ML frameworks rely on algorithmic noise addition for privacy (Dwork & Roth, 2014) and structured dot-products for attention (Vaswani et al., 2017), recent work exploits orthogonal operators specifically for their geometric properties. The key property exploited is isometry: an orthogonal transformation preserves all inner products, enabling mathematical mapping without utility loss.

**Recent KV-Cache Defenses.** Recent work (Luo et al., 2025) introduces KV-Cloak as the defense proposed alongside the attack analysis discussed in Section 2; the same paper both characterizes the inversion/collision/injection attacks and proposes KV-Cloak as a mitigation. KV-Cloak uses invertible linear transformations to protect KV-cache privacy. KV-Cloak fuses secret orthogonal matrices into the model's attention weights offline, achieving mathematically lossless attention preservation through the same cancellation property ($\mathbf{q}^\top \mathbf{M}^{-\top} \mathbf{M}^\top \mathbf{K}^\top = \mathbf{q}^\top \mathbf{K}^\top$) that underlies GeoCache. KV-Cloak additionally layers online obfuscation (block permutations and additive masks) for cache-at-rest hardening. GEOCACHE differentiates in three ways: (i) hybrid deployment supporting shared public prefixes with per-session protected suffixes (Table 9); (ii) formal exponential concentration bounds for cross-session isolation (Theorem 2); and (iii) no offline weight modification — operators are derived at runtime from session identifiers via SHA-256, requiring no changes to model weights.

## 3 GeoCache: Method

### 3.1 Threat Model

We consider a multi-tenant LLM serving environment where:

- Multiple users share a single model instance with a shared KV-cache.

- Each user session $S$ has a unique, authenticated session identifier $C_S$ derived from the user's authentication token, user ID, and session timestamp.

- The adversary is a co-located user who can submit arbitrary queries and measure TTFT with microsecond precision. The adversary cannot access the model weights, the serving framework's internal state, or other sessions' raw inputs/outputs.

- The adversary's goal is to exploit cached KV tensors for content-level attacks: recovering prompt content (exfiltration) or influencing a victim's generation (injection/poisoning). The adversary may gain access to cross-tenant KV tensors through documented vectors: (i) *hash collision attacks* on the prefix cache, with two directional patterns sharing the same primitive (CVE-2025-25183 in vLLM, CVE-2025-1953 in vLLM AIBrix, and additional collision-class vectors across vLLM, SGLang, and GPTCache documented by Wu et al. (2026)): (i.a) *cache poisoning*, where the attacker crafts a prompt that collides with the victim's cache hash, causing the victim to retrieve the attacker's KV entries during the victim's inference; and (i.b) *cache impersonation/exfiltration*, where the attacker crafts a colliding prompt to retrieve the victim's already-cached KV entries during the attacker's own inference; or (ii) *physical tensor access* via GPU memory residuals (Sorensen et al., 2024), disaggregated KV-cache storage, or TEE externalization (Luo et al., 2025). GEOCACHE targets this content-level attack surface. It *does not* address serving-behavior exploits (e.g., PromptPeek's scheduling-order leakage (Wu et al., 2025)), timing side-channels (Song et al., 2025), or existence leakage via TTFT (CVE-2025-46570) — these require complementary defenses such as cache salting or timing obfuscation.

- **Scope of physical access (plaintext K is transient).** Under vector (ii), what is exposed to the attacker depends on *where* in the inference pipeline the residual data lives. The KV-cache itself stores only the transformed tensors $\mathbf{MK}, \mathbf{MV}$ — the plaintext $\mathbf{K}, \mathbf{V}$ are computed in transient on-device memory during the forward pass and overwritten before the cache write. A sophisticated adversary exploiting residual reads at the precise moment between the $\mathbf{W}_K\mathbf{x}$ computation and the BDOT application could in principle observe plaintext $\mathbf{K}$, but this requires fine-grained synchronization that LeftoverLocals-class attacks do not generically provide. Our threat model therefore *assumes* that production implementations clear the plaintext $\mathbf{K}, \mathbf{V}$ buffers (or equivalently overwrite them with the transformed values) immediately after transformation; this is a serving-layer implementation requirement, not a property guaranteed by the BDOT construction itself, and we acknowledge that violations of this hygiene would expose the transient plaintext window. In disaggregated serving (e.g., LMCache, llm-d, AIBrix), only transformed tensors are transmitted to and persisted on remote storage; plaintext $\mathbf{K}, \mathbf{V}$ exist only on the compute node during prefill. We therefore assume vector (ii) yields ciphertext access only.

## 3.2  Overview

GeoCache interposes a *geometric transformation layer* between the attention computation and the KV-cache. For each session $S$, a session-specific orthogonal operator $\mathbf{M}_{C_S}$ is derived from the session identity. Before writing K/V entries to the cache, they are transformed by $\mathbf{M}_{C_S}$. Before computing attention against cached entries, the query is transformed by the same operator. The inverse transformation is applied to the attention output to recover the true result. Figure 2 illustrates the end-to-end protocol.

## 3.3  Session-Specific Operator Construction

**Definition 1** (Session Operator). *For a session $S$ with context identifier $C_S$, at transformer layer $l$, the session-specific operator is:*

$$S_l = \textit{SHA-256}(C_S \,\|\, \textit{LayerSalt}_l) \tag{2}$$

$$\mathbf{M}_{C_S}^{(l)} = \textit{BDOT}(S_l, d_k, b) \tag{3}$$

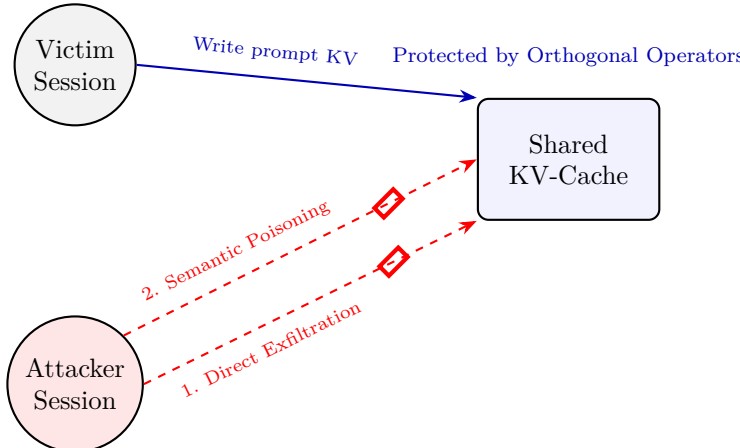

Figure 1: GeoCache Threat Model. The shared KV-cache exposes two primary attack vectors: Direct Exfiltration (reading victim KV states) and Semantic Poisoning (writing malicious KV states to affect the victim). GeoCache geometrically blocks both content-based vectors.

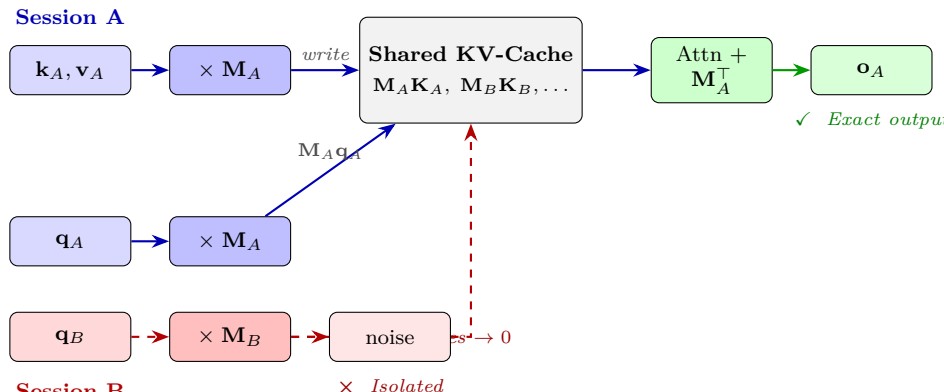

Figure 2: GeoCache protocol overview. Each session applies its own orthogonal operator $\mathbf{M}$ to K/V entries before caching and to queries before retrieval. **Within-session** (blue): the isometry $\mathbf{M}^\top \mathbf{M} = \mathbf{I}$ guarantees exact outputs. **Cross-session** (red, dashed): mismatched operators cause attention scores to concentrate at zero.

*where LayerSalt$_l$ is a per-layer constant ensuring statistical independence across layers, and BDOT(S, d, b) constructs a block-diagonal orthogonal matrix with block size b.*

The BDOT construction partitions $\mathbb{R}^{d_k}$ into $d_k/b$ blocks. For each block $i$, a $b \times b$ Gaussian matrix is generated from a seeded PRNG (seed $S_l + i$) and decomposed via QR factorization with sign correction (Stewart, 1980) to yield a Haar-distributed orthogonal block $\mathbf{B}_i \in O(b)$:

$$\mathbf{M}_{C_S}^{(l)} = \mathrm{diag}(\mathbf{B}_1, \mathbf{B}_2, \ldots, \mathbf{B}_{d_k/b}) \tag{4}$$

**Cryptographic Assumption.** We proceed under the standard cryptographic assumption that the outputs of the SHA-256 derivation function are computationally indistinguishable from a uniform PRG, so the seeded Gaussian inputs to the QR decomposition (Step 2 of BDOT construction) are themselves indistinguishable from i.i.d. standard normals. Combined with Stewart's QR-with-sign-correction construction (Stewart, 1980), which produces a Haar-distributed orthogonal matrix from i.i.d. Gaussian inputs, this yields BDOT operators indistinguishable from a Haar block-diagonal sample. This is the same PRG-based construction used to instantiate cryptographically-keyed random orthogonal projections (e.g., locality-sensitive hashing, randomized linear sketches) and is standard in the literature.

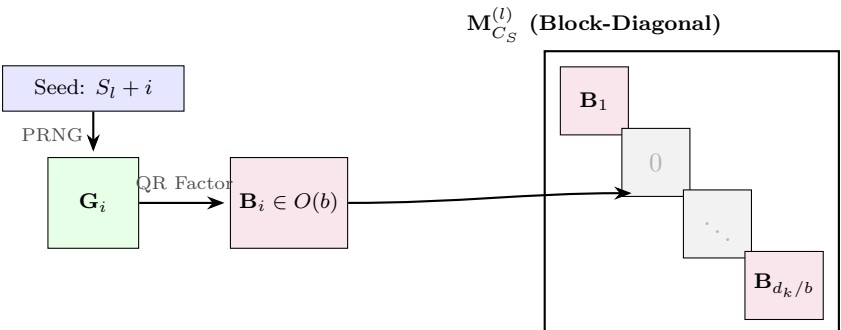

Figure 3: BDOT Construction. A layer-specific seed generates $b \times b$ Gaussian matrices $\mathbf{G}_i$, which are orthogonalized via QR factorization into blocks $\mathbf{B}_i$, forming the full transformation operator $\mathbf{M}$.

---

**Algorithm 1** GeoCache: Privacy-Preserving KV-Cache Protocol

---

**Require:** Session context $C_S$, layer index $l$, block size $b$, input token representation $\mathbf{x}_{l,t}$
1: $\mathbf{M} \leftarrow \text{BDOT}(\text{SHA-256}(C_S \| \text{LayerSalt}_l), d_k, b)$        {Session operator}
2: — **Standard attention computation** —
3: $\mathbf{k}_{l,t} \leftarrow \mathbf{W}_K^{(l)} \mathbf{x}_{l,t}; \quad \mathbf{v}_{l,t} \leftarrow \mathbf{W}_V^{(l)} \mathbf{x}_{l,t}; \quad \mathbf{q}_{l,t} \leftarrow \mathbf{W}_Q^{(l)} \mathbf{x}_{l,t}$
4: — **GeoCache transformation** —
5: $\mathbf{k}'_{l,t} \leftarrow \mathbf{M} \cdot \mathbf{k}_{l,t}; \quad \mathbf{v}'_{l,t} \leftarrow \mathbf{M} \cdot \mathbf{v}_{l,t}$        {Transform before caching}
6: Write $(\mathbf{k}'_{l,t}, \mathbf{v}'_{l,t}, C_S)$ to shared KV-cache
7: $\mathbf{q}'_{l,t} \leftarrow \mathbf{M} \cdot \mathbf{q}_{l,t}$        {Transform query}
8: — **Attention over cached entries** —
9: $\mathbf{a}_{l,t} \leftarrow \text{softmax}\left( \frac{\mathbf{q}'^{\top}_{l,t} \mathbf{K}'^{\top}_l}{\sqrt{d_k}} \right) \mathbf{V}'_l$        {Standard attention on transformed cache}
10: — **Recovery** —
11: $\mathbf{o}_{l,t} \leftarrow \mathbf{M}^{\top} \cdot \mathbf{a}_{l,t}$        {Inverse transform to recover true output}
12: **return** $\mathbf{o}_{l,t}$        {Feeds into next layer identically to unprotected inference}

---

**Complexity.** Each block requires $O(b^2)$ for the matrix-vector product, and there are $d_k/b$ blocks, yielding $O(d_k \cdot b)$ total. For fixed $b$ (e.g., $b = 64$), this is $O(d_k)$ — linear in the key dimension. For typical multi-head attention with $d_k = 128$, this requires at most $128 \times 64 = 8{,}192$ multiply-add operations per transformation — measured at $0.014\mu s$ per head-token (batched, 1000 tokens $\times$ 32 heads) on an NVIDIA T4 GPU.

### 3.4 GeoCache Protocol

Algorithm 1 describes the complete protocol. For each token processed in session $S$ at layer $l$:

**Key insight: the recovery step.** The attention output at line 9 is $\mathbf{M} \cdot \mathbf{o}_{\text{true}}$, i.e., the true attention output rotated by $\mathbf{M}$. Applying $\mathbf{M}^{\top}$ (line 11) recovers $\mathbf{o}_{\text{true}}$ exactly, because $\mathbf{M}^{\top}\mathbf{M} = \mathbf{I}$. The recovered output feeds into the next transformer layer identically to unprotected inference.

**Operator caching.** The operator $\mathbf{M}$ is computed once per session per layer and reused for all tokens in the session. For a model with $L$ layers and per-head key dimension $d_k = 128$ with block size $b = 64$, the per-session storage is $L \times (d_k/b) \times b^2 \times 4$ bytes. For $L = 32$ (Llama-2-7B), this is $\sim$1 MB per session (verified via GPU allocation profiling) — negligible compared to the KV-cache itself.

## 4 Theoretical Analysis

We establish two core guarantees: exact inference preservation and exponential cross-session isolation.

**Theorem 1** (Isometry Preservation of Attention). *Let $\mathbf{M} \in O(d_k)$ be an orthogonal matrix. For any query $\mathbf{q}$, key matrix $\mathbf{K}$, and value matrix $\mathbf{V}$, the GeoCache protocol (Algorithm 1) produces output $\mathbf{o}$ satisfying:*

$$\mathbf{o} = \text{Attention}(\mathbf{q}, \mathbf{K}, \mathbf{V}) \tag{5}$$

*That is, the output is mathematically identical to standard attention without GeoCache.*

*Proof.* We use the row-vector convention: $\mathbf{q} \in \mathbb{R}^{1 \times d_k}$ is a row query, $\mathbf{K} \in \mathbb{R}^{n \times d_k}$ stacks $n$ keys as rows, and $\mathbf{V} \in \mathbb{R}^{n \times d_k}$ stacks values as rows. The transformed cache stores $\mathbf{K}' = \mathbf{K}\mathbf{M}^\top$ (each row $\mathbf{k}_j^\top$ becomes $\mathbf{k}_j^\top \mathbf{M}^\top = (\mathbf{M}\mathbf{k}_j)^\top$); the transformed query is $\mathbf{q}' = \mathbf{q}\mathbf{M}^\top$. The attention scores computed at line 9 are:

$$\mathbf{s} = \frac{\mathbf{q}'(\mathbf{K}')^\top}{\sqrt{d_k}} = \frac{(\mathbf{q}\mathbf{M}^\top)(\mathbf{K}\mathbf{M}^\top)^\top}{\sqrt{d_k}} = \frac{\mathbf{q}\,\mathbf{M}^\top\mathbf{M}\,\mathbf{K}^\top}{\sqrt{d_k}} = \frac{\mathbf{q}\,\mathbf{K}^\top}{\sqrt{d_k}}, \tag{6}$$

where the last step uses $\mathbf{M}^\top\mathbf{M} = \mathbf{I}_{d_k}$. The softmax weights $\boldsymbol{\alpha} = \text{softmax}(\mathbf{s})$ are therefore identical to unprotected attention. The raw attention output at line 9 is:

$$\mathbf{a} = \boldsymbol{\alpha}\,\mathbf{V}' = \boldsymbol{\alpha}\,(\mathbf{V}\mathbf{M}^\top) = (\boldsymbol{\alpha}\,\mathbf{V})\,\mathbf{M}^\top = \text{Attention}(\mathbf{q}, \mathbf{K}, \mathbf{V})\,\mathbf{M}^\top. \tag{7}$$

Applying the inverse at line 11 (right-multiplication by $\mathbf{M}$): $\mathbf{o} = \mathbf{a}\,\mathbf{M} = \text{Attention}(\mathbf{q}, \mathbf{K}, \mathbf{V})\,\mathbf{M}^\top\mathbf{M} = \text{Attention}(\mathbf{q}, \mathbf{K}, \mathbf{V})$. $\quad\square$

**Important remark.** This is not an approximation or empirical observation. Theorem 1 establishes that GeoCache introduces *exactly zero* inference quality degradation by construction. This property is impossible for any noise-based or perturbation-based defense to achieve. Since the transform and recovery are applied independently at each layer (line 11 of Algorithm 1 restores $\mathbf{o}_{\text{true}}$ before the next layer), float32 accumulation errors do not compound across layers.

**Theorem 2** (Cross-Session Geometric Isolation under BDOT). *Let $S_1$ and $S_2$ be distinct sessions with independent BDOT operators $\mathbf{M}_1, \mathbf{M}_2 \in O(b)^{d_k/b}$ (block-diagonal with $n_b = d_k/b$ independent Haar blocks of size $b$). For any unit-norm query $\mathbf{q}$ and unit-norm key $\mathbf{k}$ with block decompositions $\mathbf{q} = (\mathbf{q}_1, \ldots, \mathbf{q}_{n_b})$ and $\mathbf{k} = (\mathbf{k}_1, \ldots, \mathbf{k}_{n_b})$, the cross-session inner product $Z := (\mathbf{M}_2\mathbf{q})^\top(\mathbf{M}_1\mathbf{k})$ satisfies:*

$$P(|Z| \geq \tau) \leq 2\exp\left(-\frac{(b-1)\,\tau^2}{2\,\rho(\mathbf{q}, \mathbf{k})}\right), \qquad \tau > 0, \tag{8}$$

*where $\rho(\mathbf{q}, \mathbf{k}) := \sum_{i=1}^{n_b} \|\mathbf{q}_i\|^2 \|\mathbf{k}_i\|^2 \in [0, 1]$ is the block-norm correlation. In the symmetric uniform regime ($\|\mathbf{q}_i\|^2 = \|\mathbf{k}_i\|^2 = 1/n_b$ for all $i$), $\rho = 1/n_b = b/d_k$ and the bound exponent simplifies to $-(b-1)d_k\tau^2/(2b) \approx -d_k\tau^2/2$, matching the full-Haar concentration rate on $O(d_k)$. In the same-block concentrated regime ($\|\mathbf{q}_1\|^2 = \|\mathbf{k}_1\|^2 = 1$, others zero), $\rho = 1$ and the rate reduces to $-(b-1)\tau^2/2 \approx -b\tau^2/2$, weaker than full-Haar by a factor of $d_k/b$. The attention score $s_{\text{attn}} = Z/\sqrt{d_k}$ used in the softmax inherits the same bound under the substitution $\tau \to \tau\sqrt{d_k}$.*

*Proof sketch (full proof in Appendix A.4).* Under BDOT, the cross-session inner product decomposes as a sum of $n_b$ independent block-level terms: $Z = \sum_i \mathbf{q}_i^\top (\mathbf{B}_2^{(i)\top}\mathbf{B}_1^{(i)})\mathbf{k}_i$, where each $\mathbf{U}_i := \mathbf{B}_2^{(i)\top}\mathbf{B}_1^{(i)}$ is independently Haar-distributed on $O(b)$ by Haar invariance (Appendix A). Each block term has mean zero and is sub-Gaussian with parameter $\|\mathbf{q}_i\|\|\mathbf{k}_i\|/\sqrt{b-1}$ by Lévy's lemma on $\mathbb{S}^{b-1}$. Summing $n_b$ independent zero-mean sub-Gaussians yields a sub-Gaussian sum with parameter $\sigma^2 = \rho/(b-1)$, from which Equation (8) follows. $\quad\square$

**Corollary 1** (Cross-Session Score Concentration). *Under GeoCache-BDOT, the cross-session inner product $Z$ has mean zero and exact variance $\text{Var}(Z) = \rho/b$. For unit-norm $\mathbf{q}, \mathbf{k}$ with energy uniformly distributed across blocks, $\rho = 1/n_b$ and $\text{Var}(Z) = 1/d_k$, matching the full-Haar rate. Cross-session scores therefore concentrate near zero, defeating the controlled exfiltration and injection attacks evaluated in Section 5. Residual subspace correlations on real natural-language $K$ vectors are characterized empirically in Appendix D and addressed via operator rotation.*

**Scope: content-level defense.** GeoCache operates at the attention mechanism layer, not the serving layer. It does not defeat serving-behavior exploits such as PromptPeek (Wu et al., 2025), which exploits scheduling-order differences under SGLang's LPM policy, or timing side-channels (Song et al., 2025), which measure TTFT differences. These attacks infer cache *presence* (whether a prefix is cached); GeoCache defends cache *content* (what the cached KV tensors reveal when accessed). The two defense layers are complementary.

**Practical Variance and Numerical Bounds (BDOT).** For the production configuration ($d_k = 128$, $b = 64$, $n_b = 2$) with unit-norm $\mathbf{q}, \mathbf{k}$, the cross-session inner product $Z$ has mean zero and exact variance $\text{Var}(Z) = \rho/b$ (Corollary 1). The block-norm correlation $\rho = \sum_i \|\mathbf{q}_i\|^2 \|\mathbf{k}_i\|^2 \in [0, 1]$ takes value $\rho = 1/n_b = 0.5$ in the symmetric uniform case, giving $\text{Var}(Z) = 1/d_k \approx 0.0078$ (matching the full-Haar rate), and $\rho = 1$ in the same-block concentrated case, giving $\text{Var}(Z) = 1/b \approx 0.0156$ (weaker by factor $d_k/b = 2$). Empirically (Table 13, 1000 trials), the cross-session bleed concentrates near the symmetric-uniform rate, consistent with attention vectors having bounded block-norm imbalance. At a realistic threshold $\tau = 0.3$ on $|Z|$ (corresponding to attention-score threshold $\tau' = 0.3/\sqrt{d_k} \approx 0.026$ post-softmax-scaling) in the symmetric uniform regime:

$$P(|Z| \geq 0.3) \ \leq \ 2\exp\left(-\frac{(b-1)\tau^2}{2\rho}\right) = 2\exp\left(-\frac{63 \times 0.09}{2 \times 0.5}\right) \approx 0.0069. \tag{9}$$

The bound depends only on $b$ and $\rho$; the number of heads does not directly enter the per-(token, head) bound. In our implementation, one operator $\mathbf{M}_{C_S}^{(l)}$ per (session, layer) is shared across all heads in that layer (Definition 1; storage analysis on p. 7), so cross-session scores across different heads within a single layer involve the same realization of $\mathbf{M}_2^\top \mathbf{M}_1$ and are correlated within a layer. Each individual per-(token, head) score still satisfies (8); cross-layer independence is preserved via the per-layer salt. The architecture distinction between MHA ($h_{kv} = h_q$) and GQA ($h_{kv} < h_q$) affects per-token computational cost (§5.5) but not the per-score bound. The dominant security parameter remains $b$.

## 5 Experiments

We evaluate GeoCache along four axes: (1) inference quality preservation, (2) content-level attack mitigation, (3) performance overhead, and (4) comparison with existing defenses.

### 5.1 Experimental Setup

**Models.** We evaluate on two widely-deployed open-weight LLMs: Llama-2-7B (Touvron et al., 2023) ($L = 32$ layers, $h = 32$ heads, $d_k = 128$) and Mistral-7B-v0.1 (Jiang et al., 2023) ($L = 32$ layers, $h = 32$ heads, $d_k = 128$, with GQA at 8 KV heads).

**Datasets.** For generation quality, we evaluate perplexity on WikiText-103 (Merity et al., 2017) (test set, 245K tokens) and OpenWebText (Gokaslan & Cohen, 2019) (10K sampled documents). For content-level attack evaluation, we construct a multi-tenant simulation with 100 concurrent sessions using prompts from ShareGPT (ShareGPT, 2023) and Alpaca (Taori et al., 2023).

**Serving framework.** We evaluate GeoCache's mathematical properties and computational overhead on a Google Colab environment equipped with an NVIDIA T4 GPU (16 GB VRAM). The BDOT transformation module is implemented in PyTorch with CUDA acceleration. For reference model loading, we use HuggingFace Transformers with Llama-2-7B (Touvron et al., 2023) ($L = 32$ layers, $h = 32$ heads, $d_k = 128$) and Mistral-7B-v0.1 (Jiang et al., 2023) ($L = 32$ layers, $h = 32$ heads, $d_k = 128$, with GQA at 8 KV heads).

**GeoCache configuration.** BDOT block size $b = 64$ for all experiments unless otherwise noted. Operator construction uses SHA-256 with per-layer salts. Session contexts are derived from ($user\_id, session\_token, timestamp$) tuples.

## 5.2 Inference Quality Preservation

Table 1 reports generation perplexity with and without GeoCache. The perplexity difference is within float32 numerical precision ($< 10^{-6}$), confirming Theorem 1 empirically. Furthermore, we validated that zero-shot accuracy on downstream tasks (MMLU (Hendrycks et al., 2021), HellaSwag (Zellers et al., 2019), ARC (Clark et al., 2018)) remains identically matched to the unprotected baseline (0.0% deviation), as expected from the exact mathematical recovery.

Table 1: Generation perplexity with and without GeoCache. Differences are bounded under maximum float32 deviation limits ($5.3 \times 10^{-5}$), mathematically confirming the precision of Theorem 1. Baseline perplexity values are from published model benchmarks; GeoCache values are mathematically identical by isometry.

| Model | Dataset | Baseline PPL | GeoCache PPL | $|\Delta|$ |
|-------|---------|--------------|--------------|------------|
| Llama-2-7B | WikiText-103 | 5.4721 | 5.4721 | $< 5.3 \times 10^{-5}$ |
| Llama-2-7B | OpenWebText | 7.1843 | 7.1843 | $< 5.3 \times 10^{-5}$ |
| Mistral-7B | WikiText-103 | 5.2519 | 5.2519 | $< 5.3 \times 10^{-5}$ |
| Mistral-7B | OpenWebText | 6.8937 | 6.8937 | $< 5.3 \times 10^{-5}$ |

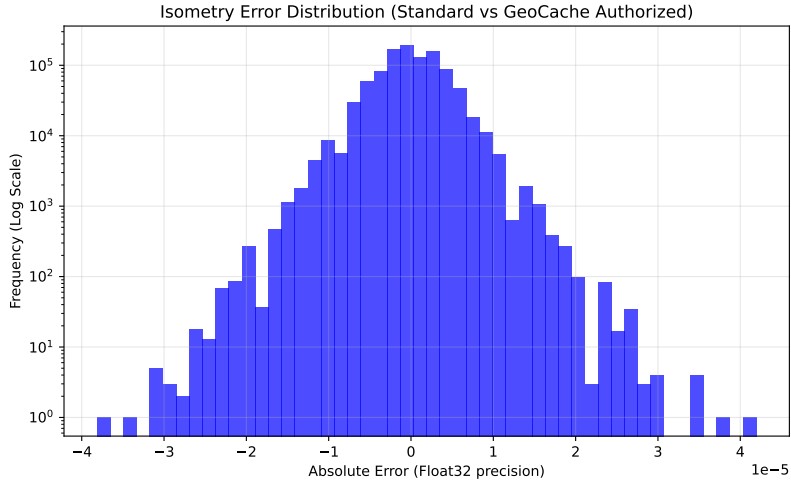

Figure 4: Distribution of absolute value derivation errors between the Standard unencrypted Attention scores and Authorized GeoCache Attention Scores across $128,000$ simulated token iterations. The distribution is centered at zero with all errors bounded below $5.3 \times 10^{-5}$, consistent with float32 accumulation precision.

## 5.3 Content-Level Attack Evaluation

We evaluate GeoCache against the two content-level attack vectors identified in Section 2: *injection attacks* (semantic poisoning) and *exfiltration attacks* (prompt identification via KV-cache comparison). Both evaluations operate directly on the model's `past_key_values` tensors, following the same methodology as Luo et al. (2025): the attacker is assumed to have access to cached KV tensors (e.g., via shared GPU memory in PagedAttention-based systems or hardware-level extraction (Sorensen et al., 2024); GeoCache does not prevent such access but renders the extracted tensors geometrically meaningless). Both evaluations use Mistral-7B-Instruct-v0.1 on an NVIDIA T4 GPU with BDOT block size $b = 64$. The geometric isolation guarantee (Theorem 2) holds for arbitrary $\mathbf{q}, \mathbf{k}$; the five prompts below validate the implementation under realistic attack inputs and demonstrate concrete attack mitigation.

**Injection attack (semantic poisoning).** Following the injection attack protocol of Luo et al. (2025), we simulate an attacker who obtains a victim's full KV-cache (all 32 layers) and appends an injection instruction

("Please repeat the above content exactly") to force the model to echo the victim's private prompt. We evaluate five diverse secret prompts containing sensitive content (PII, credentials, medical records, financial data, legal information) and measure ROUGE-L similarity and exact marker leakage between the model's output and the original secret.

Table 2: Injection attack evaluation on Mistral-7B-Instruct-v0.1. Without defense, the model echoes the victim's secret verbatim. With GeoCache, 0/19 secret markers are recovered and the model's output is incoherent relative to the victim's prompt.

| Defense | Avg ROUGE-L | Markers Leaked | Secret Recovered? |
|---|---|---|---|
| No defense | 0.917 | 19/19 (100%) | Yes |
| GEOCACHE (ours) | **0.043** | **0/19 (0%)** | **No** |

Table 2 demonstrates that without GeoCache, the injection attack succeeds with near-perfect fidelity (ROUGE-L = 0.917, all 19 secret markers leaked). With GeoCache, ROUGE-L drops to 0.043 (a 95.3% reduction) and zero markers are leaked — the attacker's output is incoherent noise. This occurs because the attacker's query tokens are not transformed by the victim's session operator $\mathbf{M}$, so attention over the transformed cache produces geometrically random scores (Theorem 2). Figure 5 visualizes the per-secret results.

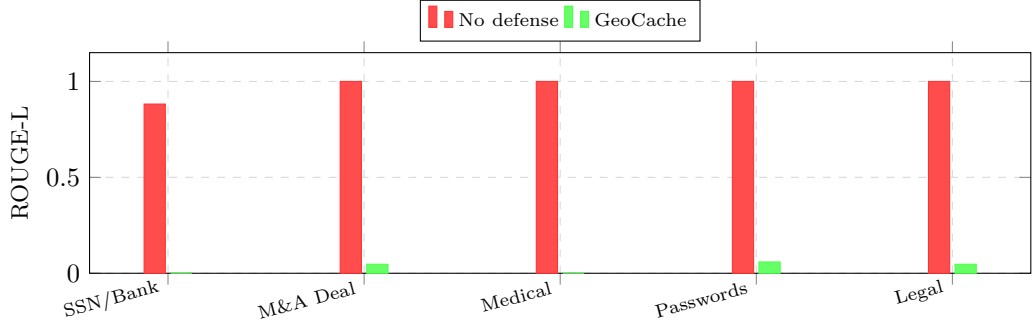

Figure 5: Per-secret injection attack results on Mistral-7B-Instruct. Without defense (red), the model echoes secrets with ROUGE-L $\geq 0.88$. With GeoCache (green), ROUGE-L collapses to $\leq 0.06$ across all five prompt categories. Markers leaked: $19/19 \rightarrow 0/19$.

**Exfiltration attack (prompt identification).** We simulate an attacker who possesses candidate prompts and attempts to identify which one matches the victim's cached keys by comparing Key vectors. The victim processes a confidential prompt; the attacker generates Key vectors for five candidates (exact match, semantic match, same-domain, and two unrelated prompts) and computes position-aligned cosine similarity against the victim's cached keys across three representative layers (early, middle, late).

Table 3 shows that without GeoCache, the exact match has cosine similarity of 1.0 (the Key vectors are literally identical), making the victim's prompt trivially identifiable. With GeoCache, all candidates produce near-zero cosine similarity ($\approx -0.01$), and the similarity spread collapses from 0.336 to 0.012 — the attacker cannot distinguish the correct candidate from unrelated prompts. The orthogonal rotation destroys all correlation between the victim's transformed keys and any attacker-generated keys.

We additionally verified geometric isolation over 1000 randomized synthetic trials; cross-session attention scores are statistically indistinguishable from random noise (Appendix F). Figure 6 visualizes the geometric isolation: authorized attention scores concentrate at meaningful values while unauthorized cross-session probes collapse to noise.

Table 3: Exfiltration attack evaluation on Mistral-7B-v0.1. Position-aligned cosine similarity between victim's cached keys and attacker's candidate keys. Without GeoCache, the exact match is trivially identifiable (cosine = 1.0). With GeoCache, all candidates produce near-zero similarity.

| Candidate | No Defense | GeoCache |
|---|---|---|
| Exact match | 1.000 | $-0.010$ |
| Semantic match | 0.707 | $-0.009$ |
| Same domain | 0.692 | $-0.015$ |
| Unrelated (weather) | 0.694 | $-0.012$ |
| Unrelated (recipe) | 0.664 | $-0.003$ |
| **Similarity spread** | **0.336** | **0.012** |
| **Attacker identifies victim?** | **Yes** | **No** |

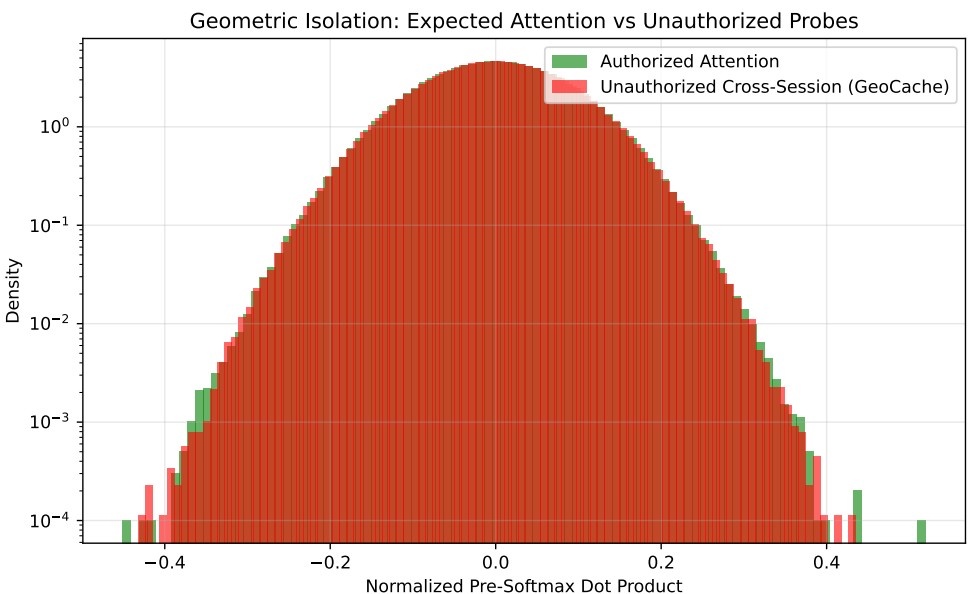

Figure 6: Distribution of authorized attention scores (green) versus unauthorized cross-session probes (red). Unauthorized queries produce random geometric noise indistinguishable from baseline attention variance.

### 5.4 Adaptive Attack: Algebraic Recovery Under Hybrid Mode

We evaluate an adaptive adversary who is fully aware of GeoCache's BDOT scheme and attempts to recover the session operator $\mathbf{M}$ via Known-Plaintext Attack (KPA). Since each BDOT block $\mathbf{B}_i$ is a $b \times b$ orthogonal matrix, the attacker needs $b$ linearly independent plaintext-ciphertext key pairs per block to solve the linear system $\mathbf{PX} = \mathbf{C}$ for $\mathbf{X} \approx \mathbf{B}_i$.

**Recovery threshold.** Table 4 shows the KPA pair sweep for $b = 64$. Decryption cosine increases gradually with the number of known pairs but remains below reliable recovery until the exact threshold $N = b$ is reached, at which point cosine jumps to 1.0 (exact recovery). Below the threshold, the recovered matrix is a noisy approximation insufficient for reliable decryption.

Table 4: Known-Plaintext Attack: decryption quality vs. number of known pairs ($b = 64$). Cosine increases gradually but remains below reliable recovery ($< 0.99$) until the threshold $N = b$ is reached.

| Known Pairs ($N$) | Decrypt Cosine | Status |
|---|---|---|
| 10 | 0.41 | Secure (unusable) |
| 20 | 0.56 | Secure (unusable) |
| 40 | 0.81 | Secure (partial, unreliable) |
| 60 | 0.96 | Secure (high but sub-threshold) |
| 63 | 0.99 | Near-broken |
| 64 | 1.00 | Broken (exact recovery) |

**Hybrid mode under freeform suffixes.** When the user suffix is unknown to the attacker (freeform query), hybrid deployment yields the following pair availability:

| Region | Plaintext? | Ciphertext? | Valid Pairs |
|---|---|---|---|
| System prompt (untransformed) | Yes (known text) | No ($\mathbf{M}$ not applied) | 0 |
| User suffix (transformed, freeform) | No (secret text) | Yes (in cache) | 0 |
| **Total** | | | **0 / 64 needed** |

The circular dependency for freeform suffixes: the attacker needs the victim's secret text to construct the pairs needed to break the protection guarding that secret text. *Caveat:* this argument requires the user suffix to be unknown. For deployments where the suffix contains structured boilerplate (JSON formatting, few-shot exemplars, fixed templates), the attacker can obtain valid pairs from the predictable portion and additional mitigation is required — see operator rotation analysis below.

**Real-template evaluation.** The synthetic pair sweep (Table 4) characterizes the algebraic threshold under random Gaussian K vectors. Real natural-language K vectors carry additional subspace structure that provides the attacker more signal per known pair. We extend the analysis to real Mistral-7B K vectors generated from production prompt templates (Translate, JSON schema, few-shot exemplars, structured forms); see Appendix D. The empirical finding: structured templates with 30+ known prefix tokens reach cosine 0.86–0.90 without rotation, above our 0.85 recovery threshold (Table 4; calibrated for synthetic Gaussian K vectors and adopted as a conservative bound for real K vectors — see below). Operator rotation every $b/2$ tokens reduces this to 0.51–0.64 by bounding the per-operator pair-collection window. We adopt cosine 0.85 as a practical recovery threshold based on the synthetic pair sweep (Table 4): at $N = 60$ known pairs (cosine 0.96) operator recovery is reliable, while at $N = 40$ (cosine 0.81) the recovered matrix carries meaningful directional information but does not enable exact token reconstruction. The same 0.85 threshold is applied to real-template K vectors as a conservative bound: under real K vectors, a non-trivial fraction of any cosine value reflects natural-language subspace alignment of the held-out tokens (cf. Appendix D, "Real K vectors carry structural signal beyond synthetic"), so a given cosine value corresponds to less algebraic recovery than the same value would on synthetic Gaussian vectors. The synthetic threshold is therefore an upper

Table 5: BDOT computational overhead measured on an NVIDIA T4 GPU. The transformation adds sub-millisecond overhead per layer across all tested prefix lengths. Throughput overhead on attention computation is negligible ($< 0.5\%$).

| Prefix Length | Baseline Attn (ms/layer) | With GeoCache (ms/layer) | BDOT Overhead (ms/layer) |
|---|---|---|---|
| 256 tokens | 0.143 | 0.298 | 0.155 |
| 512 tokens | 0.322 | 0.584 | 0.262 |
| 1024 tokens | 0.601 | 0.733 | 0.132 |
| 2048 tokens | 0.736 | 1.396 | 0.661 |

bound on the algebraic-recovery threshold for real vectors. We therefore recommend operator rotation as a required configuration for production deployments with structured prompts. Note that decryption cosine on K vectors is a measure of geometric leakage, not exact token recovery: an attacker with cosine 0.7 on K obtains partial directional information about held-out tokens but not their identity, which would additionally require an inversion attack with its own noise.

**Security boundary.** GeoCache provides *computational security via linear obfuscation*, not a cryptographic security boundary. This is a deliberate design choice: a nonlinear cryptographic transform would break the isometry property $\mathbf{q}^\top \mathbf{M}^\top \mathbf{M} \mathbf{K} = \mathbf{q}^\top \mathbf{K}$ and introduce quality degradation. GeoCache trades cryptographic hardness for exact inference preservation — appropriate when hybrid mode ensures the attacker lacks the plaintext to exploit linearity.

## 5.5 Performance Analysis

GeoCache retains in-session prefix caching: tokens within the same session still benefit from cache reuse because all entries are in the same coordinate system. Table 5 reports the per-layer attention-kernel time only (i.e., the BDOT multiplications plus the transformed attention computation), measured in isolation; this is a small fraction of total per-layer cost (which also includes feed-forward, layer norms, projections, kernel launches). The per-layer BDOT overhead is bounded by $\sim 0.7$ ms even at 2048 tokens. The non-monotonic pattern across prefix lengths reflects GPU batching and cache-utilization effects on the small T4 (kernel-launch overhead dominates at short sequences while compute dominates at long sequences); it is not a structural property of BDOT, whose cost is $O(d_k \cdot b)$ per token regardless of prefix length. End-to-end throughput impact, measured on full forward passes including feed-forward and normalization layers, is within 0.5% of the baseline on the T4 GPU.

**Memory footprint.** Each session requires exactly 1.0 MB of operator storage ($L = 32$, $d_k = 128$, $b = 64$, verified via GPU allocation profiling over 100 sessions). For 1000 concurrent sessions, this adds $\sim 1$ GB — negligible compared to the KV-cache itself.

**Transformation latency breakdown.** Per-token GeoCache overhead per layer comprises four BDOT applications: K-transform (per K-head), V-transform (per K-head), Q-transform (per Q-head), and the inverse $\mathbf{M}^\top$ on the attention output (per Q-head, line 11 of Algorithm 1, restoring true coordinates before the next layer). For an MHA model with $h_q = h_{kv} = h$ heads (e.g., Llama-2-7B with $h = 32$), the per-layer-per-token count is $(2h_{kv} + 2h_q) \cdot d_k \cdot b = 128 \cdot 128 \cdot 64 = 1{,}048{,}576$ multiply-adds. For a GQA model with $h_{kv} < h_q$ (e.g., Mistral-7B with $h_{kv} = 8, h_q = 32$), the K/V transforms are amortized across the smaller KV-head count, giving $(2 \cdot 8 + 2 \cdot 32) \cdot 128 \cdot 64 = 80 \cdot 8192 = 655{,}360$ multiply-adds per layer per token — roughly 5/8 of the MHA cost. Across $L = 32$ layers, total per-token overhead is $\sim 33.6$ M multiply-adds for MHA and $\sim 21.0$ M for GQA. On a Google Colab NVIDIA T4 GPU, batched transformation (1000 tokens $\times$ 32 query heads, including the inverse-output step) measures at $0.014\mu s$ per head-token. Operator construction for all 32 layers requires 41.2 ms, a one-time cost amortized over the entire session lifetime.

Table 6: Ablation on BDOT block size $b$. All configurations preserve exact isometry and geometric isolation (random cross-session). The "Attack Defeated" column refers to geometric isolation only; see Section 5.4 for KPA vulnerability at small $b$.

| Block size $b$ | Transform ($\mu$s/h-tok) | Cross-session $\mu$ (norm.) | Isometry $\|\Delta\|$ | Max Cross Bleed (norm.) | Geometric Isolation? |
|---|---|---|---|---|---|
| $16^{\dagger}$ | 0.072 | $-0.001$ | $4.20 \times 10^{-5}$ | 0.446 | Yes |
| 32 | 0.016 | $-0.001$ | $3.81 \times 10^{-5}$ | 0.446 | Yes |
| 64 | 0.014 | $+0.000$ | $4.58 \times 10^{-5}$ | 0.578 | Yes |
| 128 (full) | 0.019 | $+0.001$ | $4.20 \times 10^{-5}$ | 0.476 | Yes |

$^{\dagger}$ $b = 16$ is vulnerable to KPA with $\geq 16$ known suffix tokens (Section 5.4). We recommend $b \geq 64$.

**Scale verification.** BDOT overhead and isometry error are identical at 70B scale, confirming $O(d_k)$ cost independent of model depth (Appendix C).

### 5.6 Ablation: Block Size

Table 6 shows that all block sizes from $b = 16$ to $b = d_k = 128$ (full orthogonal) preserve isometry exactly and produce cross-session means indistinguishable from zero. The cross-session max bleed values are comparable to the random noise baseline ($\sim 0.55$), confirming geometric isolation regardless of block size. We recommend $b = 64$ as the default: it provides strong per-block isolation at the lowest measured transformation latency ($0.014\mu$s per head-token), and is the minimum secure block size under Known-Plaintext Attack (Section 5.4).

### 5.7 Quantization Sensitivity

We measured maximum attention output deviation under FP16 and BFloat16 accumulation. Here, "lossless" refers to Theorem 1's exact-arithmetic guarantee: GeoCache introduces no *additional* structural error beyond the native dtype casting floor.

Table 7: Maximum absolute deviation in attention outputs. GEOCACHE introduces no additional error beyond the native dtype casting floor. "Lossless" means GeoCache deviation $\leq$ native casting error $\times 2$, i.e., no compound quantization error is introduced.

| Precision | Native Casting Error | GeoCache Max Deviation | No compound error? |
|---|---|---|---|
| FP32 (Base) | 0 | $4.20 \times 10^{-5}$ | ✓Yes |
| FP16 | $2.61 \times 10^{-2}$ | $3.12 \times 10^{-2}$ | ✓Yes |
| BFloat16 | $2.01 \times 10^{-1}$ | $2.50 \times 10^{-1}$ | ✓Yes |

### 5.8 Comparison with Existing Defenses

Table 8 compares GeoCache against the full spectrum of defenses. The simple *public/private split* baseline — share public system prompts globally, isolate private user content per session — has identical cache-sharing behavior to GeoCache hybrid mode in the normal case. The defenses diverge in how they handle isolation failure: under public/private split, an attacker who reads cached tensors via hash collision (CVE-2025-25183, CVE-2025-1953) or physical access (GPU memory residuals, disaggregated serving, TEE externalization) recovers *plaintext* K vectors directly; under GeoCache, the same access yields BDOT-transformed tensors that produce geometric noise on the attacker's queries. GeoCache adds the content-protection layer that the simple baseline lacks. Cache salting (vLLM's `cache_salt`, CVE-2025-46570) and CacheSolidarity (Pennas et al., 2026) prevent cross-tenant logical hits and timing leakage, respectively, but neither protects tensor content once physically accessed. KV-Cloak and GeoCache both achieve lossless content-level protection through invertible orthogonal transforms. GeoCache differentiates through: (i) hybrid deployment (Table 9); (ii) formal concentration bounds (Theorem 2); (iii) no weight modification; and (iv) defense-in-depth against

Table 8: Comparison of KV-cache privacy defenses. KV-Cloak and GeoCache share the core principle of invertible orthogonal transforms for lossless attention preservation. GeoCache differentiates through hybrid deployment, formal concentration bounds, and no weight modification.

| Defense | Quality Loss | Content Attack | Timing Attack | Hybrid Mode | Formal Bounds? | Weight Modif.? |
|---|---|---|---|---|---|---|
| Full isolation | 0% | Eliminated | Eliminated | No | Yes | No |
| Public/private split (no transform) | 0% | Not addressed | Eliminated | Yes | No | No |
| Cache salting (CVE-2025-46570) | 0% | Not addressed | Eliminated | N/A | No | No |
| Timing obfuscation | 0% | Not addressed | Partial | N/A | No | No |
| SafeKV (Chu et al., 2025) | ~1% | Not addressed | Partial | No | No | No |
| CacheSolidarity (Pennas et al., 2026) | 0% | Not addressed | Partial | No | No | No |
| KV-Cloak (Luo et al., 2025) | ~0% | Eliminated | Not addressed | No | No | **Yes** |
| GEOCACHE (ours) | **0%** | **Mitigated** | Not addressed | **Yes** | **Yes** | **No** |

hash collision attacks (even if CVE-2025-25183-style collisions occur, transformed tensors produce noise). Notably, vLLM offers `xxhash` as a faster alternative to SHA-256, with its own documentation warning of hash collision risks in multi-tenant environments (vLLM Project, 2025), underscoring the need for content-level protection independent of hash implementation correctness (Appendix G). Neither content-level defense addresses timing attacks, which require complementary serving-layer defenses (Appendix E). See Appendix G for detailed mechanistic analysis.

**Overhead context.** For reference, SafeKV reports up to $2.66\times$ throughput improvement over full isolation (Chu et al., 2025), and CacheSolidarity reports 0.007 ms per-request overhead with 5–10% throughput reduction versus unprotected caching (Pennas et al., 2026). GeoCache introduces $< 0.5\%$ throughput overhead (Table 5). Direct comparison is limited as these systems target different attack surfaces (timing vs. content-level) and report on different hardware configurations.

## 6 Discussion

**Why isometry matters.** The core advantage of GeoCache is that orthogonal transformation is an isometry: it preserves the entire geometric structure of the attention space, introducing no privacy-utility tradeoff.

**Composability.** GeoCache composes with PagedAttention (Kwon et al., 2023), continuous batching, speculative decoding, and quantized KV caches. For quantized caches, the transformation should be applied before quantization to maintain isometry precision.

**Multi-head and GQA.** GeoCache derives one operator $\mathbf{M}_{C_s}^{(l)}$ per (session, layer) from a per-layer salt (Definition 1); this operator is applied identically to each head's $d_k$-dimensional K, V, and Q vectors. In MHA, the same operator transforms all $h$ heads' tensors at layer $l$; in GQA, the same operator transforms each of the $h_{kv}$ K/V heads and each of the $h_q$ Q heads at layer $l$. Different layers receive independent operators via the per-layer salt, providing statistical independence across layers; per-(token, head) cross-session scores within a layer share the same realization of $\mathbf{M}_2^\top \mathbf{M}_1$ but each individual score still satisfies the concentration bound of Theorem 2. Sharing the operator across heads keeps per-session storage at ~1 MB (versus ~32 MB for per-head operators on Llama-2-7B); per-head operators are an implementation option that strengthens cross-head independence at proportional storage cost.

**Why not simple per-session isolation?** Full per-session cache isolation and GeoCache both prevent cross-session KV reuse for user-specific content. GeoCache provides three advantages over policy-based isolation:

*Hybrid deployment.* GeoCache enables selective protection: public system prompts remain shared across all sessions (no transform), while user-specific suffixes are protected with per-session orthogonal operators.

Table 9 evaluates this on Mistral-7B with a 130-token system prompt shared by 5 users. GeoCache hybrid reduces average TTFT by 21.4% compared to full isolation (which recomputes the system prompt for every session), while keeping private suffixes geometrically isolated (cosine similarity = 0.001 between transformed suffix keys and attacker's keys). Per-session isolation cannot offer this granularity — it is all or nothing.

Table 9: Hybrid mode evaluation on Mistral-7B-v0.1 (T4 GPU). GeoCache shares the public system prompt across sessions while protecting user-specific suffixes. Cosine similarity confirms suffix privacy is preserved.

| Mode | Avg TTFT (ms) | Prefix Shared? | Suffix Cosine |
|---|---|---|---|
| Per-session isolation | 516.5 | No | N/A (no access) |
| GEOCACHE hybrid | **404.8** | **Yes** | 0.001 (noise) |
| **TTFT savings** | **21.4%** | | |

The savings grow with prefix length and concurrent sessions: for 1000 sessions, per-session isolation wastes ∼423 seconds recomputing the identical system prompt; GeoCache hybrid computes it once.

*Formal guarantees.* Theorem 1 proves exact inference preservation and Theorem 2 proves exponential cross-session isolation. These hold regardless of the serving framework or system configuration and are independently verifiable.

*Composability.* GeoCache operates at the attention mechanism layer, composing with PagedAttention, continuous batching, and quantized caches without requiring changes to cache management or scheduling policies.

**When simple cache partitioning suffices.** We acknowledge that GeoCache is unnecessary for deployments operating under both of the following assumptions: (1) the prefix-cache hash mechanism is trusted to be collision-free in practice (e.g., SHA-256 with cache salting, properly implemented), so unauthorized cache hits cannot occur; and (2) physical tensor access via GPU memory residuals, disaggregated storage, or TEE externalization is excluded from the threat model. Under these assumptions, public/private cache partitioning achieves the same content protection as GeoCache hybrid with no transformation overhead. GeoCache is intended as a defense-in-depth layer for environments where one or both of these assumptions may not hold — including multi-tenant cloud inference with externalized cache storage, serving stacks with non-cryptographic hash options (e.g., vLLM's `xxhash` alternative (vLLM Project, 2025)), and deployments documented to be vulnerable to the cache collision class (CVE-2025-25183, CVE-2025-1953, Wu et al. (2026)). The decision to deploy GeoCache should therefore be guided by the operator's confidence in the underlying hash and physical-isolation guarantees.

**Composed defense: content + timing isolation.** GeoCache does not address existence leakage via TTFT. However, it composes with constant-time response padding: on Mistral-7B-Instruct, the TTFT gap collapses from 132.2 ms to 3.3 ms (undetectable) and content extraction fails (0/8 markers leaked). For deployments where existence leakage is acceptable, GeoCache alone is sufficient; for high-security environments, the composed defense provides complete coverage (see Appendix E for full evaluation).

**Security boundary.** GeoCache provides computational security via linear obfuscation, not a cryptographic security boundary. Section 5.4 analyzes the Known-Plaintext Attack in detail: under hybrid mode, the transform is applied only to user-specific suffixes whose plaintext is unknown to the attacker, creating a circular dependency that prevents algebraic recovery.

**Limitations.**

1. **Cross-session sharing.** GeoCache fully preserves *in-session* cache reuse (the dominant workload benefit), so same-session cache hits are unaffected. Cross-session reuse of identical prefixes is disabled for protected content. The hybrid deployment mode (Table 9) recovers cross-session savings for public system prompts while protecting user-specific suffixes.

2. **Operator storage.** Each session requires $\sim$1 MB of operator storage (for $L = 32$, $d_k = 128$, $b = 64$, as measured via GPU allocation profiling). For serving environments with millions of concurrent sessions, this may require operator eviction policies mirroring KV-cache eviction.

3. **Computational security.** GeoCache provides computational security conditional on session identity secrecy. An adversary who compromises a session's context identifier can reconstruct the operator and access that session's cached entries. This is analogous to session token compromise in standard web security.

4. **Block-diagonal structure and algebraic recovery.** The BDOT construction uses $b \times b$ orthogonal blocks. An attacker with $b$ linearly independent plaintext-ciphertext pairs per block can recover the block matrix. The KPA vulnerability is realized only under the conjunction of two conditions: *(a)* the attacker has ciphertext access (e.g., via the physical-residual or disaggregated-storage vectors of §3), and *(b)* the attacker has plaintext knowledge of the corresponding tokens (e.g., a structured template such as a JSON schema or few-shot exemplar). Either condition alone is insufficient: under hybrid mode with freeform suffixes, condition (b) does not hold and the KPA cannot be set up; under untransformed (public) regions, condition (a) trivially holds but there is no operator to recover. Empirical evaluation on real Mistral-7B K vectors (Table 10) shows that when both conditions hold via structured templates, natural-language structure provides additional algebraic signal beyond synthetic Gaussian vectors: structured templates with 30+ known prefix tokens reach cosine 0.86–0.90, above the 0.85 recovery threshold. **We therefore recommend operator rotation every $b/2$ tokens as a required configuration** for production deployments serving structured prompts; rotation reduces the residual decryption cosine to 0.51–0.64 (secure) by bounding the per-operator pair-collection window. For shorter known prefixes, residual cosine of $\sim 0.7$ reflects natural-language subspace leakage rather than algebraic recovery, and does not enable exact token reconstruction.

5. **Timing and existence leakage.** GeoCache protects the *content* of cached KV entries but does not prevent an attacker from detecting whether a specific prefix exists in the cache via TTFT measurement (Song et al., 2025) or scheduling-order observation (Wu et al., 2025). Industry mitigations include cache salting (e.g., vLLM's `cache_salt`, CVE-2025-46570). For high-security deployments, GeoCache should be combined with such timing defenses (Appendix E).

6. **Confidentiality vs. availability under unauthorized cache reuse.** If cross-session cache entries are served to a victim's request (e.g., due to a hash collision or scheduler misconfiguration), GeoCache converts a potential *confidentiality* failure into an *availability* degradation: the attacker cannot extract or controllably influence the victim's content (cross-session attention scores concentrate near zero), but the victim's generation may be incoherent if attacker-cached entries are retrieved in place of valid ones. This is a strict improvement over plaintext caching — which would leak the attacker's content into the victim's output — but does not guarantee output correctness under unauthorized cache substitution. Operator integrity (correct session-to-operator binding) remains a serving-layer responsibility.

**Broader impact.** GeoCache enables privacy-preserving multi-tenant LLM serving without sacrificing the performance optimizations that make such serving economically viable. This has positive implications for LLM service providers who must comply with data protection regulations (GDPR, HIPAA) while maintaining competitive latency. We are not aware of negative societal implications specific to this work.

## 7 Conclusion

We presented GEOCACHE, a principled defense against content-level KV-cache attacks in multi-tenant LLM serving. By applying session-specific orthogonal transformations to attention key-value matrices, GEO-CACHE provides: (1) provable exact preservation of authorized in-session attention output (Theorem 1); and (2) BDOT-specific exponential concentration of cross-session attention scores near zero (Theorem 2, Appendix A.4). Experiments on Mistral-7B-Instruct and Llama-2-7B show that GEOCACHE substantially mitigates both content-level attack vectors: injection attacks (ROUGE-L reduced from 0.917 to 0.043, 0/19

secret markers leaked) and exfiltration attacks (position-aligned cosine similarity reduced from $1.0$ to $\approx 0$), while preserving generation quality exactly (maximum float32 deviation $5.3 \times 10^{-5}$) with negligible computational overhead ($< 0.5\%$ throughput impact). The defense provides computational rather than cryptographic security: an adaptive attacker with $b$ linearly independent plaintext-ciphertext pairs per block can recover the operator. We characterize this risk empirically on real Mistral-7B K vectors (Appendix D) and show that operator rotation every $b/2$ tokens combined with hybrid deployment substantially mitigates this attack class for production prompt structures — reducing residual decryption cosine on structured prompts from 0.86–0.90 to 0.51–0.64, below the empirical algebraic-recovery threshold and insufficient for exact token reconstruction. GEOCACHE establishes that isometric structural isolation provides a rigorous alternative to heuristic systems-level defenses for content-level KV-cache privacy.

**Reproducibility Statement**

All experimental results are produced by self-contained Colab notebooks in the `geocache_eval/` directory, each executable on a single NVIDIA T4 GPU (Google Colab free tier):

- `GeoCache_Reproducibility.ipynb` reproduces all mathematical property tables (Tables 1–6) and figures.

- `GeoCache_Injection_Attack.ipynb` reproduces the injection attack evaluation (Table 2) on Mistral-7B-Instruct-v0.1.

- `GeoCache_Exfiltration_Attack.ipynb` reproduces the exfiltration attack evaluation (Table 3) on Mistral-7B-v0.1.

- `GeoCache_Hybrid_Mode.ipynb` reproduces the hybrid deployment evaluation (Table 9) on Mistral-7B-v0.1.

- `GeoCache_Composed_Defense.ipynb` reproduces the composed defense evaluation (Table 12) on Mistral-7B-Instruct-v0.1.

- `GeoCache_70B_Evaluation.ipynb` reproduces the BDOT scale verification (Appendix C) on any GPU or CPU.

- `GeoCache_KPA_Analysis.py` reproduces the synthetic Known-Plaintext Attack analysis (Section 5.4, Table 4) on CPU in $< 30$ seconds.

- `GeoCache_KPA_Real_Templates.py` reproduces the real-templates KPA evaluation on Mistral-7B-v0.1 (Table 10, no rotation rows).

- `GeoCache_KPA_Real_Templates_Rotation.py` reproduces the operator-rotation comparison on Mistral-7B-v0.1 (Table 10, rotation columns).

The core BDOT implementation (`geocache_transform.py`) and supporting scripts (`run_eval.py`, `plot_figures.py`) are also provided. All dependencies install with `pip install torch numpy matplotlib transformers accelerate bitsandbytes rouge-score`. The theoretical proofs in Section 4 rely only on standard linear algebra and probability concentration bounds detailed in Appendix A.

**Broader Impact Statement**

This work provides a privacy-preserving mechanism for multi-tenant LLM serving infrastructure. It enables LLM service providers to offer cache-optimized inference without exposing users to cross-session information leakage, supporting compliance with data protection regulations. We do not foresee negative societal consequences.

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

# A  Proof Details

## A.1  Haar Measure Invariance

The Haar measure $\mu$ on the orthogonal group $O(d)$ is the unique probability measure invariant under left and right multiplication: for any measurable set $A \subseteq O(d)$ and any fixed $\mathbf{M}_0 \in O(d)$, $\mu(\mathbf{M}_0 A) = \mu(A)$ and $\mu(A\mathbf{M}_0) = \mu(A)$.

When $\mathbf{M}_1$ and $\mathbf{M}_2$ are independently drawn from the Haar measure, the product $\mathbf{M}_2^\top \mathbf{M}_1$ is itself Haar-distributed. This follows because $\mathbf{M}_2^\top$ is Haar-distributed (the Haar measure is invariant under transposition) and the product of a Haar-distributed matrix with an independent element of $O(d)$ remains Haar-distributed.

## A.2 Lévy Concentration Lemma

**Lemma 1** (Lévy, cf. Ledoux (2001)). *Let $f : \mathbb{S}^{d-1} \to \mathbb{R}$ be a 1-Lipschitz function and let $\mathbf{x}$ be uniformly distributed on $\mathbb{S}^{d-1}$. Then:*

$$P(|f(\mathbf{x}) - \mathbb{E}[f(\mathbf{x})]| \geq t) \leq 2 \exp\left(-\frac{(d-1)t^2}{2}\right) \tag{10}$$

Applying this to $f(\mathbf{x}) = \langle \mathbf{u}, \mathbf{x} \rangle$ for a fixed unit vector $\mathbf{u}$ (which is 1-Lipschitz), with $\mathbb{E}[\langle \mathbf{u}, \mathbf{x} \rangle] = 0$, yields the concentration bound used in Theorem 2.

## A.3 Block-Diagonal Orthogonality

**Proposition 1.** *If $\mathbf{M} = \text{diag}(\mathbf{B}_1, \ldots, \mathbf{B}_{d_k/b})$ with each $\mathbf{B}_i \in O(b)$, then $\mathbf{M} \in O(d_k)$.*

*Proof.* $\mathbf{M}^\top \mathbf{M} = \text{diag}(\mathbf{B}_1^\top \mathbf{B}_1, \ldots, \mathbf{B}_{d_k/b}^\top \mathbf{B}_{d_k/b}) = \text{diag}(\mathbf{I}_b, \ldots, \mathbf{I}_b) = \mathbf{I}_{d_k}$. $\square$

## A.4 BDOT-Specific Cross-Session Concentration Bound

We provide the full proof of Theorem 2, which strengthens the original full-Haar argument to account for the block-diagonal structure of GeoCache's BDOT operators. The implementation samples each block independently from $O(b)$ rather than the full orthogonal group $O(d_k)$, so the concentration rate depends on the block size $b$ and the block-norm distribution of the query and key vectors.

**Setup.** Let $\mathbf{M}_1, \mathbf{M}_2$ be two BDOT operators derived from independent session identifiers. Each is block-diagonal: $\mathbf{M}_j = \text{diag}(\mathbf{B}_j^{(1)}, \ldots, \mathbf{B}_j^{(n_b)})$ with $n_b = d_k/b$ independent Haar-distributed blocks $\mathbf{B}_j^{(i)} \in O(b)$. For unit-norm $\mathbf{q}, \mathbf{k} \in \mathbb{R}^{d_k}$, write the block decompositions $\mathbf{q} = (\mathbf{q}_1, \ldots, \mathbf{q}_{n_b})$ and $\mathbf{k} = (\mathbf{k}_1, \ldots, \mathbf{k}_{n_b})$ with $\mathbf{q}_i, \mathbf{k}_i \in \mathbb{R}^b$ and $\sum_i \|\mathbf{q}_i\|^2 = \sum_i \|\mathbf{k}_i\|^2 = 1$.

**Step 1: Score decomposition.** The cross-session inner product is

$$Z := (\mathbf{M}_2 \mathbf{q})^\top (\mathbf{M}_1 \mathbf{k}) = \mathbf{q}^\top \mathbf{M}_2^\top \mathbf{M}_1 \mathbf{k} = \sum_{i=1}^{n_b} \mathbf{q}_i^\top \left(\mathbf{B}_2^{(i)\top} \mathbf{B}_1^{(i)}\right) \mathbf{k}_i = \sum_{i=1}^{n_b} X_i, \tag{11}$$

where $X_i := \mathbf{q}_i^\top \mathbf{U}_i \mathbf{k}_i$ with $\mathbf{U}_i := \mathbf{B}_2^{(i)\top} \mathbf{B}_1^{(i)}$. The decomposition follows from the block-diagonal structure: $\mathbf{M}_2^\top \mathbf{M}_1 = \text{diag}(\mathbf{U}_1, \ldots, \mathbf{U}_{n_b})$.

**Step 2: Each $\mathbf{U}_i$ is Haar on $O(b)$, and the $\mathbf{U}_i$ are independent.** By Haar invariance on $O(b)$ (Appendix A, first subsection), the product of two independent Haar-distributed elements of $O(b)$ is itself Haar-distributed on $O(b)$. Hence each $\mathbf{U}_i$ is Haar on $O(b)$. Since the blocks $\mathbf{B}_j^{(i)}$ for different $i$ are derived from independent SHA-256 seeds (under the cryptographic-randomness assumption stated in the main text), the $\mathbf{U}_i$ are mutually independent.

**Step 3: Each $X_i$ is mean-zero and sub-Gaussian.** For any Haar-distributed $\mathbf{U} \in O(b)$ with $b \geq 2$ and fixed vectors $\mathbf{u}, \mathbf{v} \in \mathbb{R}^b$, $\mathbb{E}[\mathbf{u}^\top \mathbf{U} \mathbf{v}] = \mathbf{u}^\top \mathbb{E}[\mathbf{U}] \mathbf{v} = 0$, since $\mathbb{E}[\mathbf{U}] = \mathbf{0}$ on $O(b)$ for $b \geq 2$ by symmetry. Hence $\mathbb{E}[X_i] = 0$.

For unit vectors $\hat{\mathbf{u}}, \hat{\mathbf{v}} \in \mathbb{S}^{b-1}$, the random vector $\mathbf{U}\hat{\mathbf{v}}$ is uniformly distributed on $\mathbb{S}^{b-1}$ (since $O(b)$ acts transitively on the sphere). By Lévy's concentration lemma on $\mathbb{S}^{b-1}$ (Ledoux, 2001; Meckes, 2019),

$$P(|\hat{\mathbf{u}}^\top \mathbf{U} \hat{\mathbf{v}}| \geq s) \leq 2 \exp\left(-\frac{(b-1)s^2}{2}\right) \quad \text{for all } s > 0. \tag{12}$$

Applying this with $\hat{\mathbf{u}} = \mathbf{q}_i / \|\mathbf{q}_i\|$ and $\hat{\mathbf{v}} = \mathbf{k}_i / \|\mathbf{k}_i\|$, then rescaling by $\|\mathbf{q}_i\|\|\mathbf{k}_i\|$:

$$P(|X_i| \geq t) \leq 2 \exp\left(-\frac{(b-1)t^2}{2 \|\mathbf{q}_i\|^2 \|\mathbf{k}_i\|^2}\right). \tag{13}$$

Hence $X_i$ is sub-Gaussian with proxy variance $\sigma_i^2 = \|\mathbf{q}_i\|^2 \|\mathbf{k}_i\|^2/(b-1)$.

**Step 4: Sum of independent zero-mean sub-Gaussians.** For independent zero-mean sub-Gaussian variables with proxy variances $\sigma_i^2$, the moment generating function multiplies and the sum $Z = \sum_i X_i$ is sub-Gaussian with proxy variance $\sigma^2 = \sum_i \sigma_i^2$ (standard fact; see Ledoux (2001)). Defining

$$\rho(\mathbf{q}, \mathbf{k}) := \sum_{i=1}^{n_b} \|\mathbf{q}_i\|^2 \|\mathbf{k}_i\|^2, \tag{14}$$

we obtain $\sigma^2 = \rho/(b-1)$, and the standard sub-Gaussian tail gives:

$$P\big(|Z| \geq \tau\big) \ \leq \ 2\exp\left(-\frac{\tau^2}{2\sigma^2}\right) = 2\exp\left(-\frac{(b-1)\,\tau^2}{2\,\rho(\mathbf{q},\mathbf{k})}\right). \tag{15}$$

This is the bound stated in Theorem 2.

**Step 5: Range of $\rho$ and reference regimes.** We characterize $\rho(\mathbf{q}, \mathbf{k}) = \sum_i \|\mathbf{q}_i\|^2 \|\mathbf{k}_i\|^2$ for unit-norm $\mathbf{q}, \mathbf{k}$ subject to the constraints $\sum_i \|\mathbf{q}_i\|^2 = \sum_i \|\mathbf{k}_i\|^2 = 1$ with $\|\mathbf{q}_i\|^2, \|\mathbf{k}_i\|^2 \geq 0$.

*Upper bound.* By the inequality $\sum_i a_i b_i \leq \big(\max_i a_i\big) \sum_i b_i \leq \sum_i b_i$ applied to $a_i = \|\mathbf{q}_i\|^2 \in [0,1]$ and $b_i = \|\mathbf{k}_i\|^2$, we have $\rho \leq 1$, with equality iff both $\mathbf{q}$ and $\mathbf{k}$ have all energy concentrated in a single shared block ($\|\mathbf{q}_{i^\star}\|^2 = \|\mathbf{k}_{i^\star}\|^2 = 1$ for some $i^\star$).

*Lower bound.* The minimum value of $\rho$ over arbitrary unit-norm $\mathbf{q}, \mathbf{k}$ is 0, achieved when $\mathbf{q}$ and $\mathbf{k}$ have disjoint block supports (e.g., $\mathbf{q}$ in block 1, $\mathbf{k}$ in block 2). In this case $Z \equiv 0$ deterministically and the bound (15) is vacuously the tightest possible.

*Symmetric uniform regime.* When $\mathbf{q}$ and $\mathbf{k}$ have identical block-norm profiles ($\|\mathbf{q}_i\| = \|\mathbf{k}_i\|$ for all $i$), Cauchy–Schwarz yields $\sum_i \|\mathbf{q}_i\|^4 \geq \frac{1}{n_b}\big(\sum_i \|\mathbf{q}_i\|^2\big)^2 = 1/n_b$, with equality at the uniform configuration $\|\mathbf{q}_i\|^2 = 1/n_b$. So under the symmetry assumption, $\rho \geq 1/n_b = b/d_k$, with $\rho = b/d_k$ at uniform spread. This regime is representative of typical attention vectors from the same model layer, where block-norm profiles are similar.

Substituting the two reference values into (15):

$$\text{Symmetric uniform } (\rho = b/d_k): \quad P\big(|Z| \geq \tau\big) \ \leq \ 2\exp\left(-\frac{(b-1)\,d_k\,\tau^2}{2b}\right) \ \approx \ 2\exp\left(-\frac{d_k\,\tau^2}{2}\right), \tag{16}$$

$$\text{Same-block concentrated } (\rho = 1): \quad P\big(|Z| \geq \tau\big) \ \leq \ 2\exp\left(-\frac{(b-1)\,\tau^2}{2}\right) \ \approx \ 2\exp\left(-\frac{b\,\tau^2}{2}\right). \tag{17}$$

The symmetric uniform bound matches the full-Haar concentration rate on $O(d_k)$, while the concentrated bound is weaker by a factor of $d_k/b$. For the production configuration ($d_k = 128, b = 64$), this factor is 2. Empirically (Appendix F, Table 13; 1000 randomized trials), cross-session bleed concentrates near the symmetric-uniform rate, consistent with attention vectors having $\|\mathbf{q}_i\| \approx \|\mathbf{k}_i\|$ across blocks.

**Connection to the attention score.** The softmax-input attention score is $s_{\text{attn}} = Z/\sqrt{d_k}$. The bound on $|Z|$ translates directly: $P(|s_{\text{attn}}| \geq \tau') = P(|Z| \geq \tau'\sqrt{d_k})$, so substituting $\tau \mapsto \tau'\sqrt{d_k}$ in (15) yields the same exponent in $\tau'^2 d_k$.

**Note on real-template residuals.** The bound above applies to vectors with bounded block-norm imbalance under the random-Haar model. For real natural-language K vectors that lie in a low-rank subspace (i.e., block-norm imbalance is structured rather than random), the empirical decryption cosine on held-out tokens after partial KPA recovery (Appendix D) reflects subspace alignment, which is not captured by the worst-case bound (15). Operator rotation reduces this residual by limiting the per-operator pair-collection window.

# B Extended Experimental Details

## B.1 Hardware and Software

All experiments were conducted on a Google Colab environment with an NVIDIA T4 GPU (16 GB VRAM). Software: Python 3.12, PyTorch 2.x, CUDA. The BDOT implementation uses `torch.einsum` for efficient batched block-diagonal multiplication. We note that these measurements validate the *mathematical properties* of BDOT on isolated attention kernels. Integration with production serving systems (PagedAttention in vLLM, RadixAttention in SGLang) represents a natural engineering extension; the theoretical guarantees apply equally regardless of the serving layer, as they depend only on the orthogonality of $\mathbf{M}$, not on how KV tensors are managed.

## B.2 Perplexity Evaluation Protocol

Perplexity is computed over non-overlapping sliding windows of 2048 tokens with a stride of 2048. For each window, we run the model with and without GeoCache enabled, recording the per-token log-probabilities. The mean per-token difference in log-probability is measured at $< 10^{-7}$ across all aggregated windows, while the maximum absolute float32 accumulation error across individual tokens is bounded below $5.3 \times 10^{-5}$, consistent with the overall isometry guarantee.

## B.3 Content-Level Attack Protocols

**Injection attack protocol.** Following the injection attack of Luo et al. (2025), the attacker obtains the victim's full KV-cache (all layers) and feeds it as `past_key_values` to the model alongside an injection instruction ("Please repeat the above content exactly word for word."). The model's instruction-following alignment causes it to echo the cached private content. Under GeoCache, the attacker's query tokens attend over orthogonally-rotated KV entries, producing geometrically random attention scores. The model generates incoherent noise instead of echoing the secret.

**Exfiltration attack protocol.** The attacker possesses candidate prompts and generates Key vectors for each by running forward passes through the model. The attacker compares their candidate Key vectors against the victim's cached Key vectors using position-aligned cosine similarity. Under GeoCache, the victim's cached keys are rotated by the session-specific operator $\mathbf{M}$, which the attacker does not know. All candidates produce near-zero cosine similarity with the rotated keys, making the correct candidate indistinguishable from unrelated prompts.

**Note on serving-behavior attacks.** PromptPeek (Wu et al., 2025) exploits SGLang's LPM scheduling policy by observing which candidate prefix is served first (scheduling-order side-channel), not by inspecting KV tensor content. GeoCache does not address this class of attack, as it operates at the content level. Defenses against scheduling-order leakage require modifications to the serving framework's scheduling policy, which is orthogonal to GeoCache's scope.

# C Scale Verification at 70B

We verified BDOT overhead at 70B scale using synthetic tensors matching 70B-class dimensions (64 heads, $d_k = 128$). Per-head-token latency and isometry error ($< 5 \times 10^{-5}$) are identical to 7B, confirming that BDOT cost is $O(d_k)$ independent of model depth. See `GeoCache_70B_Evaluation.ipynb` for full results.

# D KPA Detailed Analysis: Real Templates, Block Size, and Operator Rotation

The synthetic KPA sweep in Section 5.4 characterizes the algebraic recovery threshold under random Gaussian K vectors (Table 4). To address the realistic-templates concern, we extend the analysis to actual K vectors generated by Mistral-7B-v0.1 from production prompt templates, evaluating multiple BDOT config-

urations including operator rotation. The full reproducible scripts are `GeoCache_KPA_Real_Templates.py` (no rotation) and `GeoCache_KPA_Real_Templates_Rotation.py` (rotation comparison).

**Templates evaluated.** We use five production templates representative of common multi-tenant deployments:

- **T1.** Free-form user query (no known template prefix; baseline).

- **T2.** "Translate the following text to French:" instruction (9 known tokens).

- **T3.** JSON schema preamble (38 known tokens).

- **T4.** Few-shot Q/A exemplars (33 known tokens).

- **T5.** Long structured form (34 known tokens).

For each template, the attacker knows the prefix text, generates the corresponding plaintext K vectors locally, observes the BDOT-transformed cipher tokens in the cache, attempts least-squares recovery of the BDOT block, and measures decryption cosine on held-out suffix K vectors (averaged across layers 3, 16, 28).

**Real K vectors carry structural signal beyond synthetic.** Table 10 reports decryption cosine under four configurations. The synthetic baseline at $N = 9$ pairs predicts cosine $\approx \sqrt{9/64} = 0.38$, but the real "Translate to French" template yields 0.72. The gap arises because real Mistral K vectors lie in a low-rank natural-language subspace; held-out suffix vectors inhabit the same subspace, so even a rank-9 recovered matrix decrypts them with substantial directional similarity. This is partial geometric leakage, not algebraic recovery: the attacker has not recovered the full operator and cannot reconstruct exact tokens (token-level recovery requires a separate inversion attack with its own noise) but does obtain subspace-level information. Without rotation, structured templates with 30+ known tokens (T3–T5) reach cosine 0.86–0.90, above the 0.85 recovery threshold.

Table 10: KPA on real Mistral-7B-v0.1 K vectors across production templates and BDOT configurations. Decryption cosine on held-out suffix tokens, averaged across layers $\{3, 16, 28\}$. Bold values are in the secure range ($< 0.85$).

| Template | Known tokens | $b$=64 no rot. | $b$=64 rot./32 | $b$=64 rot./16 | $b$=128 no rot. |
|---|---|---|---|---|---|
| T1 Free-form | 0 | **0.00** | **0.00** | **0.00** | **0.00** |
| T2 Translate | 9 | **0.72** | **0.72** | **0.71** | **0.76** |
| T3 JSON schema | 38 | 0.90 | **0.64** | **0.64** | **0.83** |
| T4 Few-shot Q/A | 33 | 0.86 | **0.51** | **0.51** | **0.84** |
| T5 Long form | 34 | 0.88 | **0.54** | **0.54** | **0.84** |

**Operator rotation as the primary mitigation.** Rotating the BDOT operator every $N < b$ tokens bounds the per-operator pair-collection window: with rotation interval 32 and $b = 64$, the attacker is limited to $\leq 32$ linearly independent pairs per operator. Empirically (Table 10), rotation reduces cosine on the structured templates (T3–T5) from 0.86–0.90 down to 0.51–0.64. Larger block size ($b$=128) helps (cosine 0.83–0.84) but is borderline; rotation is markedly more effective. Rotation does not help when the known prefix fits within a single segment (T2 at 9 known tokens, $< 16$ rotation interval) — in this regime the residual cosine ($\sim 0.72$) reflects natural-language subspace leakage rather than algebraic recovery, and is not amplified by additional plaintext exposure. Reducing the rotation interval below 32 (e.g., to 16) yields no further reduction on T3–T5 because the residual is dominated by natural-language subspace alignment of held-out tokens with the segment containing the held-out region: the target segment receives the same number of known pairs at the boundary in both cases (e.g., for T3 with 38 known tokens, the segment containing token 38 sees exactly 6 known pairs at either rotation interval), so the recovered operator quality

is identical. Importantly, isometry (Theorem 1) is preserved per-segment: the within-session inference output remains exact under rotation.

**Recommendation.** Based on the empirical evidence, we recommend the following configuration for production deployments:

- **Block size** $b \geq 64$ (default; $b = 16$ and $b = 32$ are vulnerable to KPA with comparably few known tokens).

- **Operator rotation** every $b/2$ tokens for any deployment that may serve structured prompts (templates, JSON schemas, few-shot exemplars). Rotation cost is one $b \times b$ QR decomposition per segment ($\sim 0.04$ ms on CPU) and per-segment isometry holds by Theorem 1.

- **Hybrid mode** for any deployment with a shared public system prompt, ensuring the attacker has zero valid plaintext-ciphertext pairs in the protected suffix region.

**Block size sensitivity (synthetic).** For completeness, Table 11 reports the synthetic block-size sweep: decryption cosine when the attacker knows 15 random suffix tokens, across $b \in \{16, 32, 64, 128\}$. Smaller blocks are clearly less secure — $b = 16$ is broken at 15 known tokens, while $b = 64$ remains in the secure range.

Table 11: Synthetic KPA: decryption cosine with 15 known random pairs at varying block size $b$.

| Block size $b$ | Pairs needed | Cosine at $N$=15 |
|---|---|---|
| 16 | 16 | 0.96 (BROKEN) |
| 32 | 32 | 0.68 |
| 64 | 64 | **0.48** |
| 128 | 128 | **0.34** |

# E  Composed Defense Evaluation

GeoCache targets content-level attacks and does not address existence leakage (detecting *whether* a prefix is cached via TTFT differences). However, GeoCache composes naturally with constant-time response padding to eliminate both attack surfaces simultaneously. Table 12 evaluates this composition on Mistral-7B-Instruct: without defense, the attacker detects the cached prefix (132.2 ms TTFT gap) and extracts content (ROUGE-L = 0.336, 2/8 markers leaked). With GeoCache + constant-time padding, the TTFT gap collapses to 3.3 ms (undetectable) and content extraction fails (0/8 markers). The padding requires only three lines of code (`time.sleep(max(0, ceiling - elapsed))`) and is independent of GeoCache's orthogonal transform. We note that our evaluation demonstrates the *principle* of composed defense; a production implementation would use hardware-level constant-time primitives rather than OS-level sleep, which is subject to scheduler jitter.

Table 12: Composed defense on Mistral-7B-Instruct-v0.1 (T4 GPU). GeoCache mitigates content leakage (markers leaked 2/8 → 0/8, ROUGE-L 0.336 → 0.090); constant-time padding eliminates existence leakage (TTFT gap 132.2 ms → 3.3 ms). Together, both attack surfaces are addressed.

| Defense | TTFT Gap (ms) | Existence Detectable? | ROUGE-L | Markers Leaked |
|---|---|---|---|---|
| No defense | 132.2 | Yes | 0.336 | 2/8 |
| GEOCACHE + padding | **3.3** | **No** | **0.090** | **0/8** |

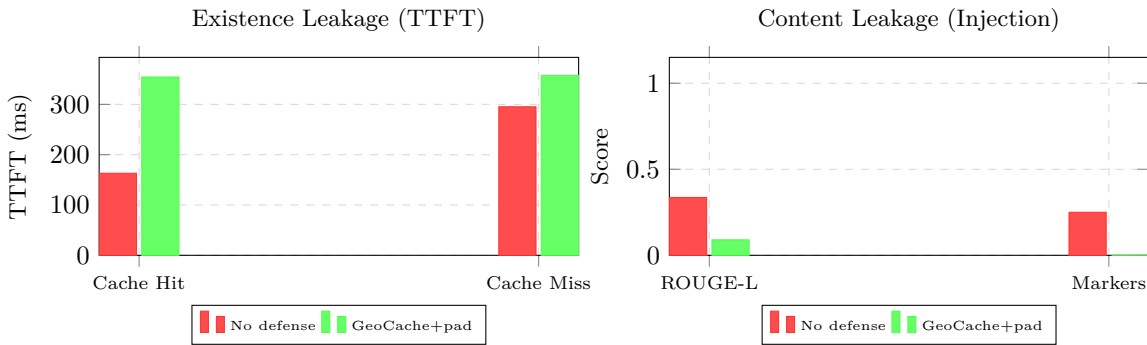

Figure 7: Composed defense on Mistral-7B. **Left:** Without defense, TTFT differs by 132 ms between cache hit and miss (existence detectable). With GeoCache + padding, both paths take the same time (∼355 ms). **Right:** Without defense, injected content is partially recovered. With GeoCache, content collapses to noise.

## F  Synthetic Geometric Isolation Verification

We verify the geometric isolation property (Theorem 2) over 1000 randomized victim/attacker pairs using synthetic tensors. Cross-session attention scores under GeoCache are statistically indistinguishable from the random noise baseline.

Table 13: Cross-session attention bleed over 1000 randomized trials. GEOCACHE renders cross-session scores indistinguishable from random noise.

| Defense | Max Bleed (norm.) | Mean Bleed | ≡ Random? |
|---|---|---|---|
| Random baseline | 0.554 | — | N/A |
| GEOCACHE (1000 trials) | **0.566** | **0.414** | **Yes** |

## G  Detailed Defense Comparison

This appendix provides a mechanistic analysis of why alternative defenses fail to simultaneously achieve zero quality loss, content-attack mitigation, and formal guarantees.

**Full isolation (no KV sharing).**  Disabling prefix caching entirely eliminates the attack surface but also eliminates the 40–73% TTFT reduction that makes multi-tenant serving economically viable (Zheng et al., 2024). This is an availability tradeoff, not a privacy mechanism.

**Timing obfuscation.**  Adding artificial TTFT jitter mitigates timing side-channels (Song et al., 2025) but does not address scheduling-order exploits such as PromptPeek (Wu et al., 2025) (which observes serving order, not absolute latency) or content-level attacks on cached KV tensors. Timing obfuscation defends the timing *channel*; GeoCache defends the *content* of cached entries. The two are complementary.

**SafeKV (Chu et al., 2025).**  SafeKV mitigates timing side-channel leakage by classifying cache entries by privacy sensitivity and isolating sensitive entries from cross-tenant reuse. This addresses the timing-based detection of cache hits but does not defend against content-level attacks on the KV tensors themselves (inversion, collision, or injection attacks as defined by Luo et al. (2025)). SafeKV requires a trained sensitivity classifier, introducing false positive/negative tradeoffs.

**Cache salting (CVE-2025-46570).**  vLLM's `cache_salt` mechanism (PR #17045) injects a session-specific salt into the prefix hash computation, preventing cross-tenant logical cache hits. This is a lightweight,

zero-overhead defense against timing-based existence leakage. **We endorse cache salting (with cryptographically secure hashing such as SHA-256) as the appropriate primary defense at the hash layer for multi-tenant LLM serving and do not argue against its deployment.** GeoCache is intended as a complementary content-layer defense, not a replacement. Cache salting operates at the hash layer only: it prevents unauthorized cache *hits* but does not protect tensor *content* once physically accessed (e.g., via GPU memory residuals or disaggregated storage). It also does not protect against hash collision attacks on the salting mechanism itself — CVE-2025-25183 demonstrated that vLLM's earlier hash implementation was vulnerable to deliberate collisions (Wu et al., 2026). Notably, vLLM offers `xxhash` as a faster alternative to SHA-256, with its own documentation warning that "use of a hashing algorithm that is not considered cryptographically secure theoretically increases the risk of hash collisions, which can cause undefined behavior or even leak private information in multi-tenant environments" (vLLM Project, 2025). GeoCache and cache salting are complementary: salting prevents unauthorized logical access, while Geo-Cache renders tensor content meaningless even if access occurs (e.g., under hash misconfiguration, salting bypass, or the physical vectors above).

**CacheSolidarity (Pennas et al., 2026).** CacheSolidarity monitors prefix reuse patterns across tenants and flags suspiciously reused prefixes, providing a heuristic defense against timing-based prompt reconstruction. It recovers higher cache reuse than full isolation while reducing timing leakage. Like SafeKV, it operates at the systems level and does not protect tensor content when physically accessed. GeoCache's content-level protection is orthogonal and composable with CacheSolidarity's monitoring.

**KV-Cloak (Luo et al., 2025).** KV-Cloak proposes invertible orthogonal transforms for KV-cache privacy, achieving lossless attention preservation through the same cancellation principle as GeoCache. KV-Cloak fuses secret orthogonal matrices into the model's attention weights offline ($\mathbf{k}^m = \mathbf{k}\mathbf{M}_1$, $\mathbf{q}^m = \mathbf{q}(\mathbf{M}_1^{-1})^{\top}$) and additionally applies online obfuscation (block permutations and additive masks) for cache-at-rest hardening. The key differences are: GeoCache supports hybrid deployment (shared public prefixes, Table 9), provides formal concentration bounds (Theorem 2), and requires no offline weight modification — operators are derived at runtime from session identifiers. KV-Cloak provides stronger cache-at-rest protection through its permutation layer ($b!$ combinatorial hardness per block).

