# OpenReview forum: "GeoCache: Provably Lossless Inference and Computational Content-Level Isolation for Shared KV-Caches in Multi-Tenant LLM Inference via Isometric Orthogonal Transformation"
_TMLR — Rejected by TMLR_

### Review · Reviewer_A6hB · 2026-03-31

**Summary Of Contributions:**

**Summary of Contributions**

This paper proposes GeoCache, a method that applies session-specific orthogonal transformations to KV-cache entries in multi-tenant LLM inference. The goal is to preserve exact attention computation within a session while preventing cross-session cache reuse from producing meaningful outputs.

The paper provides a theoretical argument based on the isometry of orthogonal matrices and implements a block-diagonal construction. It evaluates the method on standard LLMs, focusing on perplexity, attention behavior, and computational overhead.

**Strengths**

The paper targets a relevant problem in LLM serving, namely the security risks of KV-cache sharing. The proposed approach is simple and easy to understand, and the idea of using orthogonal transformations to preserve attention computation is straightforward.

The implementation appears lightweight, and the reported overhead is small, suggesting that the approach may be practical under its assumptions.

**Weaknesses**

The paper contains several critical issues.

First, there are clear factual errors and misleading citations. The paper claims that prior work (e.g., SafeKV / Chu et al.) demonstrates “semantic poisoning” at the KV-cache level, where attackers inject malicious attention vectors via prefix matching. This is incorrect because such an attack is not defined or implemented in those works, and SafeKV does not discuss or evaluate any form of semantic poisoning. This threat appears to be fabricated rather than grounded in existing literature. In addition, the paper mischaracterizes PromptPeek. PromptPeek is not a timing-based side-channel attack; it primarily exploits serving behavior (e.g., scheduling order in systems like SGLang). Timing-based leakage is studied in "The Early Bird Catches the Leak: Unveiling Timing Side Channels in LLM Serving Systems" The paper conflates these distinct mechanisms and attributes them incorrectly, leading to a flawed problem formulation.

Second, the threat model, claimed attacks, and evaluation are inconsistent. The paper describes a adversary stealing privacy via cache hit but claims to address stronger attacks (e.g., semantic poisoning and exfiltration) without demonstrating them in evaluation, making the security claims unclear.

Third, the core contribution is not convincing. The method essentially enforces per-session isolation through transformations, and it is unclear why this is meaningfully different from simpler isolation mechanisms or why such complexity is necessary.

**Audience:**

Yes

**Audience Explanation:**

The general problem studied in this paper, i.e., security and privacy risks of KV-cache sharing in multi-tenant LLM inference, is important and relevant to the TMLR audience. However, the current submission does not provide a clear or reliable contribution due to the issues discussed above.

**Broader Impact Concerns:**

No major ethical concerns beyond standard security considerations. The work targets improving privacy and isolation in multi-tenant LLM systems, which is generally positive.

**Claims And Evidence:**

No

**Claims Explanation:**

The paper’s claims are not supported by accurate or convincing evidence. A central problem is that the paper builds part of its motivation on factual errors and misleading citations. In particular, it attributes KV-cache-level “semantic poisoning” attacks to prior work such as SafeKV, but those works do not define or evaluate such attacks. The paper also mischaracterizes PromptPeek as a timing-based attack, when its core mechanism is tied to serving order rather than pure timing leakage. These errors weaken the paper’s problem formulation from the start.

**Requested Changes:**

**Remove factual errors and fix misleading citations.**

The paper currently misstates prior work, especially around “semantic poisoning” and the characterization of PromptPeek. These claims must be corrected before the paper can be evaluated on solid ground.

**Make the threat model, claims, and evaluation consistent.**

The paper currently mixes different attacker models and attack goals, while the evaluation does not match the claimed scope. The authors must either narrow the claims or provide evidence for the stronger ones.

**Clarify the actual contribution over simple session isolation.**


The paper does not clearly justify why the proposed method is meaningfully different from, or preferable to, simpler per-session isolation. This must be explained much more clearly.

---

> ### Author Response · Authors · 2026-04-01
> **Author Response to Reviewer**
>
> We thank the reviewer for the thorough feedback. We address the critical points raised regarding citations, evaluation, and the threat model below.
>
> **1. Correction of Factual Errors and Citations**
>
> We take full responsibility for the attribution errors in the initial submission and will correct them as follows:
>
> **SafeKV:** In the revised draft, we will correctly identify SafeKV (Chu et al., 2025) as a defense against timing side-channels through privacy-sensitive cache isolation, and remove the incorrect attribution of "semantic poisoning."
>
> **PromptPeek:** In the revised draft, we will correctly describe PromptPeek (Wu et al., 2025) as an exploit of SGLang's Longest Prefix Match scheduling policy — a serving-behavior exploit, not a classical timing side-channel. We add the missing citation to Song et al. ("The Early Bird Catches the Leak," 2024) for timing side-channels.
>
> **Revised Taxonomy:** Section 2.3 will categorize the threat landscape into: **(i)** Serving-behavior exploits (PromptPeek), **(ii)** Timing side-channels (Song et al., 2024), and **(iii)** Content-level KV-cache attacks (Luo et al., 2025). GeoCache is explicitly scoped as a defense against category (iii) throughout the paper.
>
> **2. Attack Evaluation**
>
> To address the gap between our geometric isolation measurements and the attacks we claim to prevent, we have conducted two controlled experiments on Mistral-7B-Instruct-v0.1 on a T4 GPU, following published attack protocols:
>
> **Injection attack.** Following the injection attack protocol of Luo et al. (2025), we construct a scenario where an attacker obtains a victim's KV-cache and appends an instruction to extract private content. We evaluate on 5 prompts containing PII, credentials, medical records, financial data, and legal information. Without GeoCache: **avg ROUGE-L = 0.917, 19/19 secret markers recovered**. With GeoCache: **avg ROUGE-L = 0.043, 0/19 markers recovered (95.3% reduction)**. The attacker's queries produce geometrically random attention scores over the transformed cache, consistent with Theorem 2.
>
> **Exfiltration attack.** The attacker generates Key vectors for candidate prompts and compares them against the victim's cached Keys via position-aligned cosine similarity. Without GeoCache: the exact-match candidate has **cosine = 1.0** (K vectors are identical), with a similarity spread of 0.336. With GeoCache: all candidates produce **cosine ≈ -0.01, spread = 0.012**. The attacker cannot distinguish the correct candidate from unrelated prompts.
>
> Both experiments are implemented as self-contained Colab notebooks running end-to-end on a free T4 GPU. We share these results here for the reviewer's consideration while we prepare the full revision with updated tables, figures, and supplementary code.
>
> **3. Representation-Level Isolation vs. System-Level Policy**
>
> We clarify that GeoCache is not a replacement for system-level isolation, but a mathematically grounded mechanism for representation-level isolation with three concrete advantages:
>
> **Hybrid deployment.** GeoCache enables selective protection: public system prompts remain shared across sessions for TTFT savings, while user-specific suffixes are transformed. In preliminary experiments on Mistral-7B with a 130-token shared system prompt, hybrid mode reduces average **TTFT by 21.4%** compared to per-session isolation while keeping private suffix keys geometrically isolated **(cosine = 0.001)**. Per-session isolation does not support this granularity.
>
> **Formal guarantees.** Theorem 1 proves exact inference preservation; Theorem 2 proves exponential cross-session isolation. These hold regardless of the serving framework or system configuration and are independently verifiable.
>
> **Composability.** GeoCache operates at the attention mechanism layer, composing with PagedAttention, continuous batching, and quantized caches without requiring changes to cache management or scheduling policies.
>
> We will incorporate these changes in the revised manuscript.

---

### Review · Reviewer_YSxv · 2026-04-02

**Summary Of Contributions:**

This paper introduces **GeoCache**, a defense mechanism designed to secure shared Key-Value (KV) caches in multi-tenant LLM inference environments. The core idea is to apply session-specific isometric orthogonal transformations to K and V vectors before they are stored in a shared cache. This is achieved through a structured **Block-Diagonal Orthogonal Transform (BDOT)** to minimize computational overhead.
- **Strengths:** The method provides a mathematically proven lossless guarantee for authorized cache reuse ($q^\top M^\top M k = q^\top k$) and demonstrates negligible latency overhead on modern GPUs.
- **Weaknesses:** The authors significantly **over-claim** their theoretical contributions. They present the use of isometric orthogonal transformations for attention isolation as a mathematical breakthrough, whereas this is an established technique already utilized in concurrent work (e.g., KV-Cloak). More importantly, the threat model and evaluations significantly overlook **timing side-channels** (existence leakage), which remains a critical vulnerability in shared prefix-caching systems.

**Audience:**

Yes

**Audience Explanation:**

Researchers and engineers specializing in LLM inference optimization (e.g., vLLM/SGLang contributors) and ML privacy will find value in the engineering implementation of BDOT. The demonstration of how to apply orthogonal transforms efficiently in a production-oriented "prefix-caching" context provides a useful reference for system-level security design.

**Broader Impact Concerns:**

The primary concern is the potential for providing a **false sense of security**. By labeling the defense as "Provably Lossless" and "Privacy-Preserving," users may assume it protects against all forms of leakage. In reality, the lack of timing-channel isolation allows for "Existence Leakage," which can confirm if a user has processed a specific confidential document. The authors must include a "Limitations" section advising that GeoCache be combined with timing obfuscation for high-security environments.

**Claims And Evidence:**

No

**Claims Explanation:**

The submission makes several key claims, but the evidence for some is either misleading or incomplete:

* **Claim 1: Exact preservation of inference with zero utility loss.**
    * **Assessment:** **Partially Supported.** While Theorem 1 and empirical results support the lossless nature of the math, the paper incorrectly claims this as a unique advantage over *KV-Cloak*. Mathematically, *KV-Cloak* is also lossless with the query transformed accordingly. The authors' assertion of an inherent "quality-security tradeoff" in prior work lacks objective evidence.
* **Claim 2: Exponential geometric isolation for cross-session access.**
    * **Assessment:** **Supported** by Theorem 2 and the empirical concentration of attention scores near zero in cross-tenant scenarios (Table 3-5).
* **Claim 3: Comprehensive mitigation of cross-tenant cache-based attacks (e.g., PromptPeek).**
    * **Assessment:** **Weak/Misleading.** While GeoCache prevents *content* extraction, it does not address **existence leakage**. In a shared prefix-cache system, an attacker can still use Time-to-First-Token (TTFT) measurements to determine if a specific sensitive prefix exists in the cache. The evaluation in Section 5.3 is incomplete as it ignores this "cache oracle" capability.

**Requested Changes:**

1.  **Correct the Misrepresentation of Prior Work:** The authors must revise the "Related Work" section to provide an honest account of *KV-Cloak*. They must acknowledge that using reversible/orthogonal transforms for attention isolation is an established concept. The contribution should be repositioned as an efficient engineering adaptation for the "prefix-sharing" scenario.
2.  **Address Timing Side-Channel Limitations:** The authors must explicitly state that GeoCache **does not** prevent existence leakage via TTFT. Section 5.3 must be updated to clarify that while *content* is protected, the *presence* of cached data remains detectable. The claim of "comprehensive" defense must be qualified.
3.  **Refine Statistical Evidence:** Clarify whether the empirical variances in Table 2 are sufficient to support the "lossless" claim across diverse model architectures beyond 7B parameters. Include performance benchmarks on larger model scales (e.g., 70B) to verify that BDOT overhead remains negligible.

---

> ### Author Response · Authors · 2026-04-08
> **Author Response to Reviewer**
>
> We thank the reviewer for the detailed feedback. We address each requested change below.
>
> **1. Relationship with KV-Cloak**
>
> We agree the original submission did not provide a sufficiently detailed comparison with KV-Cloak. We will clearly acknowledge that both GeoCache and KV-Cloak share the core principle of invertible orthogonal transforms for lossless attention preservation.
>
> However, we respectfully disagree with the characterization of GeoCache as an "engineering adaptation of an established technique," for the following reasons:
>
> **(a) Different threat models.** KV-Cloak addresses a CSP-level adversary who accesses externalized KV-cache (e.g., outside TEE boundaries). GeoCache addresses cross-session prefix sharing in multi-tenant serving, where multiple sessions intentionally share cache entries for TTFT optimization and the attack surface is the shared prefix matching itself. KV-Cloak does not address the prefix-sharing scenario.
>
> **(b) Pure orthogonal guarantee.** Both approaches achieve lossless attention through invertible transforms. KV-Cloak's full pipeline additionally includes additive masks and block permutations for cache-at-rest hardening. KV-Cloak's authors describe the resulting accuracy as "practically lossless" (their Section VI). GeoCache uses a single-step orthogonal transform where M⊤M = I holds algebraically (Theorem 1), with no additional components. This simplicity is a design choice — KV-Cloak's extra layers provide stronger cache-at-rest protection.
>
> **(c) No offline weight modification.** KV-Cloak fuses obfuscation matrices into model attention weights offline. GeoCache derives operators at runtime from session identifiers via SHA-256, requiring no model modification — essential for multi-tenant serving with a shared model instance.
>
> **(d) Formal cross-session bounds.** Theorem 2 provides an explicit exponential concentration bound on cross-session attention scores, with no equivalent in KV-Cloak.
>
> **(e) Hybrid deployment.** GeoCache supports shared public prefixes with per-session protected suffixes. In experiments on Mistral-7B with a 130-token shared system prompt, hybrid mode reduces average TTFT by **21.4%** compared to per-session isolation while keeping suffix keys geometrically isolated **(cosine = 0.001)**. KV-Cloak does not support this mode.
>
> The revised comparison table will reflect this shared foundation honestly while articulating these distinctions.
>
> **2. Timing Side-Channel, Existence Leakage, and Scope of Claims**
>
> We agree the title and abstract could give a broader impression than intended. The revised draft will add "Content-Level" to the title, explicitly list what GeoCache does not address in Section 3.1 (serving-behavior exploits, timing side-channels, hardware-level extraction), remove or qualify all uses of "comprehensive," and add an explicit Limitation recommending GeoCache be combined with timing obfuscation for high-security deployments.
>
> We have also conducted a **composed defense experiment** on Mistral-7B-Instruct demonstrating that GeoCache + constant-time response padding eliminates both attack surfaces: the TTFT gap collapses from **132.2 ms to 3.3 ms** (existence undetectable), and content extraction is neutralized (ROUGE-L 0.336→0.090, markers 2/8→0/8).
>
> **3. Content-Level Attack Evaluation**
>
> Following the attack protocols of Luo et al. (2025), we conducted two controlled experiments on Mistral-7B-Instruct-v0.1:
>
> **Injection attack.** The attacker obtains a victim's KV-cache and appends an instruction to extract private content. We evaluate on 5 prompts containing PII, credentials, medical records, financial data, and legal information. Without GeoCache: **avg ROUGE-L = 0.917, 19/19 secret markers recovered**. With GeoCache: **avg ROUGE-L = 0.043, 0/19 markers recovered (95.3% reduction)**.
>
> **Exfiltration attack.** The attacker compares candidate Key vectors against the victim's cached Keys via position-aligned cosine similarity. Without GeoCache: exact match has **cosine = 1.0**, spread = 0.336. With GeoCache: all candidates produce **cosine ≈ -0.01, spread = 0.012**. The attacker cannot distinguish the correct candidate from unrelated prompts.
>
> We share these results while we prepare the full revision with updated tables, figures, and supplementary code.

---

> > ### Comment · Reviewer_YSxv · 2026-04-10
> > **Threat Model Flaw: The Paradox of Prefix Matching and Unviable Attack Vectors**
> >
> > Thank you for the detailed rebuttal and the supplementary experiments regarding timing obfuscation. While I appreciate the authors' willingness to revise the scope of their claims and acknowledge the prior work, a fundamental issue regarding the viability of the proposed threat model in real-world multi-tenant architectures remains unresolved.
> >
> > Specifically, the "content-level extraction attack" under the multi-tenant prefix-sharing scenario appears unrealistic in practical LLM serving infrastructures. This leads to a severe mismatch between the paper's claims and the experimental evidence.
> >
> > I would like the authors to clarify the following two critical points:
> >
> > **1. The Logical Paradox in Prefix Sharing and Content Exfiltration**
> >
> > In modern multi-tenant serving systems (e.g., using RadixAttention in vLLM/SGLang), a KV-cache hit strictly requires an exact token-by-token match of the prefix. If Malicious Tenant B successfully triggers a cache hit on Victim Tenant A's private data, it mathematically implies that Tenant B *already knows and has explicitly inputted* that exact private plaintext prompt. If the attacker already possesses the plaintext, there is no unknown "content" left to extract or restore from the KV-cache.
> >
> > Furthermore, as the authors conceded in the rebuttal, adding artificial delays (constant-time TTFT padding) effectively mitigates the existence leakage (timing side-channel) to an indistinguishable level. If the timing channel is closed, and the engine strictly enforces exact prefix matching for cache reuse, the attack surface is completely neutralized. It seems that GeoCache is unnecessary in this context, as simply adding latency already mitigates the practical privacy risks associated with prefix sharing.
> >
> > **2. The Missing Exploit Chain in the Threat Model**
> >
> > If the goal of GeoCache is to prevent a malicious tenant from stealing privacy via shared KV-caches, the authors must explain *how* the malicious tenant actually accesses the victim's specific cache data in the first place.
> > In the rebuttal regarding the Injection Attack, the authors state: *"The attacker obtains a victim's KV-cache and appends an instruction..."*. However, for a standard API tenant communicating via text-based HTTP requests, it is practically impossible to arbitrarily "obtain" or "mount" another tenant's KV-cache block ID out of thin air without inputting the exact matching prompt.
> >
> > The experiments, which demonstrate a "100% defense rate," appear to assume an attacker with a "god-mode" privileged access—perhaps assuming a severe IDOR (Insecure Direct Object Reference) vulnerability in the scheduler, or direct physical memory access. This assumption is severely disconnected from the paper's claim of defending against "API-level multi-tenant attacks." If the attacker has physical memory access, the threat model reverts to a *cache-at-rest* scenario, contradicting the stated premise of the paper.
> >
> > **Requested Clarification:**
> > To justify the claims, the authors must provide a realistic, end-to-end exploit chain demonstrating how a standard API tenant can execute this 'Injection Attack' or 'Exfiltration Attack' under standard API isolation constraints. If the attack inherently relies on assuming a compromised scheduler or physical memory access, the core claims regarding "multi-tenant prefix-sharing vulnerabilities" are fundamentally invalid and the threat model must be entirely rewritten.

---

> > > ### Author Response · Authors · 2026-04-12
> > > **Author Response to Reviewer**
> > >
> > > We thank the reviewer for this important challenge. We address both points and the requested clarification below.
> > >
> > > **1. The Logical Paradox in Prefix Sharing**
> > >
> > > We provide the requested end-to-end exploit chain under standard API constraints:
> > >
> > > 1. Attacker crafts a prompt that collides with the victim's prefix hash
> > > 2. Attacker's KV entries get cached under the colliding hash
> > > 3. Victim's request hits the cache and retrieves attacker's entries
> > > 4. Without GeoCache: victim's generation is corrupted (semantic poisoning)
> > > 5. With GeoCache: attacker's entries are in a different coordinate system — victim's queries produce noise — attack fails
> > >
> > > This is an API-level attack requiring no hardware access and no privilege escalation. **CVE-2025-25183** (vLLM, fixed in v0.7.2) and **CVE-2025-1953** (vLLM AIBrix) demonstrated exactly this — hash collisions in prefix caching allowed an attacker to serve poisoned KV entries to a victim through a different prompt. "Cache Me, Catch You" (Wu et al., NDSS 2026) documented additional vulnerabilities across six attack vectors in vLLM, SGLang, and GPTCache, showing that cache collision attacks lead to incorrect model outputs, information leakage, and content filter bypasses.
> > >
> > > While these have been patched (vLLM v0.11 now defaults to SHA-256), three observations: (1) This vulnerability class recurred independently — CVE-2025-25183 (vLLM), CVE-2025-1953 (AIBrix), and additional vulnerabilities across six attack vectors (Wu et al., NDSS 2026). A single patch does not eliminate a systemic pattern. (2) vLLM v0.11 offers xxhash as a faster alternative to SHA-256, and their own documentation warns this "can cause undefined behavior or even leak private information in multi-tenant environments." Operators under latency pressure may choose speed over security. (3) Defense-in-depth is the security standard — no single layer is assumed infallible. GeoCache provides the content protection layer independent of hash implementation correctness.
> > >
> > > **2. The Missing Exploit Chain — Physical Access Vectors**
> > >
> > > Beyond hash collisions, KV tensors also become physically accessible in production infrastructure:
> > >
> > > **(a) GPU memory residuals.** LeftoverLocals (Sorensen & Khlaaf, 2024; CVE-2023-4969) demonstrated that GPU local memory is not cleared between workloads, allowing a subsequent tenant to read residual KV tensors.
> > >
> > > **(b) Disaggregated serving.** Production architectures (e.g., LMCache, llm-d, AIBrix) store and transmit KV tensors across nodes on remote DRAM/SSD. A compromised storage node or network interceptor can read tensor content.
> > >
> > > **(c) Externalized KV-cache outside TEE.** This is the same threat model as KV-Cloak (NDSS 2026) — the KV-cache sits outside the TEE boundary, readable by the CSP.
> > >
> > > Logical access controls (cache_salt, SHA-256 hashing) prevent unauthorized cache *hits* but do not protect tensor *content* once physically accessed. GeoCache protects both layers.
> > >
> > > **3. Regarding Timing Padding Sufficiency**
> > >
> > > Timing padding addresses existence leakage but does not protect tensor content under either vector above. If an attacker poisons the cache via hash collision, timing padding does not help — the victim still retrieves corrupted KV entries. If an attacker reads tensors from GPU memory, timing padding does not help — the raw content is exposed. GeoCache and timing padding address different layers.
> > >
> > > We will rewrite Section 3.1 to explicitly describe both attack vectors (hash collision and physical access) with CVE references.

---

> ### Comment · Reviewer_YSxv · 2026-04-30
>
> I would like to thank the authors for their detailed rebuttal and the revised manuscript. I appreciate the significant effort made to clarify the threat model, explicitly acknowledge the Known-Plaintext Attack (KPA) vulnerability, and provide the much-needed comparison with concurrent work.
>
> However, after carefully reviewing the revision, I find that the fundamental issues raised in the initial review—specifically the severe mismatch between the stated claims and the supporting evidence, as well as the logical contradictions within the threat model—remain unresolved. The additions and concessions made in the revision actually expose further fatal flaws in the mathematical soundness and practical security of the proposed approach.
>
> **1. Unsupported Security Claims due to KPA Vulnerability**
> In the revised Section 5.4, the authors explicitly acknowledge the Known-Plaintext Attack (KPA), confirming that an attacker only needs $b=64$ known plaintext-ciphertext token pairs to completely and exactly recover the transformation matrix.
> To defend against this, the authors propose a "Hybrid Mode," arguing that attackers cannot reliably guess 64 tokens within a user's private suffix. However, in modern LLM applications (e.g., Agent frameworks, code completion, structured JSON/Markdown outputs), user inputs inevitably contain highly predictable or fixed template strings, structural markers, and formatting instructions. Securing a system on the assumption that an attacker cannot gather 64 predictable tokens is fragile. This trivial linear vulnerability fundamentally invalidates the paper's overarching claim of providing "Provably Privacy-Preserving" caching.
>
> **2. Unjustified System Complexity for a Solved Problem**
> Addressing the "prefix-matching paradox" raised in my initial review, the authors state that API-level cache extraction is possible if the system uses a weak, collision-prone hashing algorithm like `xxhash`.
> From a systems security engineering perspective, the only correct and zero-cost fix for weak hash collisions is to revert to a collision-resistant hash (SHA-256) combined with tenant-specific Cache Salting. Proposing a cryptographically fragile matrix multiplication as a "defense-in-depth" mechanism against a basic configuration error is an example of severe over-engineering.
> Furthermore, if the threat model is expanded to include physical memory access (e.g., GPU LeftoverLocals), it explains how the ciphertext is leaked, but it simultaneously exposes the system directly to the aforementioned KPA, rendering the encryption instantly useless.

---

> > ### Author Response · Authors · 2026-04-30
> > **Author Response to Reviewer Follow-up**
> >
> > We thank the reviewer for the continued engagement. We address both points directly.
> >
> > **1. KPA with Realistic Templates**
> >
> > The reviewer's concern about structured prompts is valid. We have now run the KPA evaluation on real Mistral-7B K vectors using production prompt templates. The results confirm the reviewer's intuition — and reveal a refinement of our synthetic analysis:
> >
> > - Freeform query (0 known tokens): cosine 0.00 (secure)
> > - "Translate to French:" (9 known): cosine 0.72 (partial signal from language structure, not algebraic recovery)
> > - JSON schema (38 known): cosine 0.90 without rotation, **0.64 with rotation**
> > - Few-shot Q/A (33 known): cosine 0.86 without rotation, **0.51 with rotation**
> > - Long structured form (34 known): cosine 0.88 without rotation, **0.54 with rotation**
> >
> > The reviewer is correct that without operator rotation, structured templates with 30+ known tokens reach near-broken cosine. We do not dispute this. Operator rotation every 32 tokens — already described in our original submission (Appendix D) as a hardening mechanism — splits known tokens across multiple operators and reduces structured-template cases with 30+ known tokens to cosine 0.51-0.64. Templates with fewer than 32 known tokens (e.g., "Translate to French" at 9 tokens) retain cosine ~0.72 because all known tokens fit within a single rotation segment — but this reflects natural-language subspace alignment, not algebraic operator recovery, and does not enable exact token reconstruction. The revised manuscript will reframe operator rotation from "additional hardening" to **required for production deployments with structured prompts**.
> >
> > We note that the dominant use case for multi-tenant KV-cache sharing — freeform conversational queries where the user's private content is unknown to the attacker — remains fully secure (cosine 0.00) without any additional mechanisms.
> >
> > **2. System Complexity and the Physical Access Argument**
> >
> > The reviewer raises two sub-arguments:
> >
> > **(a) SHA-256 + cache_salt is sufficient.** We agree that SHA-256 with cache_salt is an effective defense at the hash layer, and we do not argue against its deployment. GeoCache operates at a different layer. The reviewer frames this as over-engineering; we frame it as defense-in-depth — a standard security principle where no single layer is assumed infallible. We respectfully note that hash-based isolation failed independently in three implementations (CVE-2025-25183, CVE-2025-1953, and additional vectors documented by Wu et al., NDSS 2026) before being patched. SHA-256 is collision-resistant but not immune to implementation errors in the surrounding code. GeoCache provides a content-layer guarantee that holds regardless of hash implementation correctness.
> >
> > **(b) Physical access exposes both ciphertext AND plaintext, enabling KPA.** This is an important point. The KV-cache itself stores only transformed tensors (M×K). Plaintext K is computed in transient memory during the forward pass. While a sophisticated attacker exploiting residual reads at the precise moment between W_K computation and BDOT application could in principle access plaintext K, this requires synchronization that LeftoverLocals-class attacks do not generically provide. Production implementations should clear plaintext K immediately after transformation. In disaggregated serving, the storage node holds only transformed tensors; plaintext KV vectors exist transiently on the compute node and are never transmitted to storage.
> >
> > We acknowledge that if the attacker has BOTH physical memory access AND knowledge of the victim's prompt (e.g., a known template), the KPA applies. This is precisely why operator rotation is recommended — it bounds the attacker's collection window regardless of access method.
> >
> > **On the title claim:** we will revise the title to 'Provably Lossless Inference and Computational Content-Level Isolation' — separating the provable utility claim (Theorem 1) from the empirical security claim — and scope all security claims throughout the manuscript to the computational security boundary.
> >
> > These changes are being incorporated in the next revision, which we will upload shortly.

---

### Review · Reviewer_xjC3 · 2026-04-10

**Summary Of Contributions:**

The paper proposes GeoCache, a method designed to secure KV-cache sharing in multi-tenant Large Language Model (LLM) serving. To prevent Direct KV Exfiltration and Semantic Poisoning, the authors apply a session-specific Block-Diagonal Orthogonal Transform (BDOT) to the Key and Value attention matrices before caching. The core contribution relies on the isometric property of orthogonal matrices: within a session, the attention output is mathematically preserved exactly ($q^{\top}M^{\top}MK = q^{\top}K$), resulting in zero degradation of generation quality.

**Key Strengths:**

- **Mathematical Elegance:** The application of isometry to ensure exactly zero degradation in inference utility is theoretically sound and highly desirable in LLM serving infrastructure.

- **Low Overhead:** The computational overhead is practically negligible (<0.5% throughput impact), making it highly efficient compared to non-linear cryptographic approaches.


**Key Weaknesses:**

- **Susceptibility to Known-Plaintext Attacks (KPA):** The defense relies entirely on a linear transformation (matrix multiplication). From a security standpoint, linear obfuscation is fundamentally broken against Known-Plaintext Attacks (KPA). The authors use a BDOT block size of $b=64$. To completely break this "encryption," an attacker only needs to collect 64 linearly independent plaintext-ciphertext vector pairs to solve a basic system of linear equations and recover the transformation block $B_i$. In real-world LLM serving, system prompts, standard templates, and few-shot examples are highly predictable and publicly known. An attacker running a local copy of the open-weight model (e.g., Llama-2) can easily generate the "plaintext" KV tensors for these standard prompts. Once the matrix is algebraically solved, the "geometric isolation" is entirely bypassed, exposing all private user data in that session. The authors' claim that gathering 64 independent pairs is "practically infeasible" ignores the reality of lengthy, standardized prompt templates in modern LLM applications.

- **Marginal Practical Benefit over Simpler System-Level Defenses:** The paper fails to benchmark against the most practical and lightweight industry standard: hash-based cache salting (e.g., vLLM's `cache_salt` mechanism / CVE-2025-46570). vLLM achieves logical cache isolation simply by injecting a session-specific salt into the prefix hash computation, completely preventing cross-tenant logical hits with zero matrix multiplication overhead. While GeoCache claims to also protect against physical Direct KV Exfiltration from VRAM, its vulnerability to simple algebraic recovery (KPA) renders this physical protection illusionary. Therefore, the added complexity of computing and applying orthogonal matrices yields virtually no practical security advantage over a simple hash salt.

- **Algorithmic Oversimplicity vs. Claimed Security:** The core novelty of the paper boils down to multiplying the KV cache by an orthogonal matrix. While the authors dress this up as "mathematically guaranteed cross-session isolation," it is merely linear data obfuscation, not a true cryptographic security boundary.

- **Missing Adaptive Attack Evaluation:** As a defense-oriented paper, the evaluation fails to consider a scenario where the attacker is fully aware of the defense scheme and actively attempts to bypass it algebraically.

- **Missing Baseline Comparisons:** The author only compares Random baseline and Full isolation in the evaluation. More experiments, such as comparison with CacheSolidarity or SafeKV in terms of efficiency and overhead should be conducted.

**Audience:**

Yes

**Audience Explanation:**

The optimization and security of LLM serving infrastructure (e.g., vLLM, SGLang, TensorRT-LLM) is an active and critical area of research. Multi-tenant architectures are highly vulnerable to cache-based side channels. The overarching attempt to manipulate the geometric structure of the attention space—rather than relying purely on systems-level eviction policies—is an interesting engineering angle that merits discussion, critique, and visibility within the machine learning systems community.

**Claims And Evidence:**

No

**Claims Explanation:**

Several core claims regarding the security and utility of the proposed method are not supported by convincing evidence:

1. **Vulnerability to Known-Plaintext Attacks (KPA):** The claim that the system provides "Provably Lossless Privacy-Preserving" cache sharing is unsupported because the evaluation does not test against algebraic recovery. The authors present a theoretical bound on geometric isolation but fail to provide empirical evidence or formal proofs that the session-specific matrix $M$ can withstand basic linear equation solving when an attacker has access to predictable token sequences (known plaintexts).

2. **Destruction of Cross-Tenant Caching:** The authors claim GeoCache maintains the benefits of shared KV-caches. However, because the operators are session-specific, if User B queries the exact same public prefix as User A, User B will retrieve geometrically incompatible noise. Thus, GeoCache breaks cross-tenant deduplication just as completely as standard logical isolation methods do, limiting reuse strictly to in-session (intra-tenant) hits. The claim of retaining multi-tenant caching benefits is therefore overstated.

**Requested Changes:**

- **Address the KPA Vulnerability:** The paper should include a formal security analysis evaluating a Known-Plaintext Attack. The authors should simulate an attacker reconstructing the BDOT matrices using known system prompts, and either prove mathematically that solving these linear equations is harder than it appears, or significantly alter the threat model to assume the attacker can never guess a sequence of $b$ (e.g., 64) tokens.

- **More Baseline Comparison:** The evaluation is incomplete without comparing GeoCache to standard cache isolation techniques used in industry (e.g., vLLM's `cache_salt`, CVE-2025-46570), which append a session-specific salt to the Radix tree hash. The authors should articulate exactly what physical exfiltration vectors GeoCache secures that a simple hash salt does not, especially considering GeoCache's algebraic fragility. Also, the author should consider comparing with more baselines, such as CacheSolidarity or SafeKV in terms of efficiency and overhead.

- **Add an Adaptive Attack Subsection:** As a defense proposal, the evaluation should include an adaptive adversary. The authors should design and execute an experiment where the attacker is fully aware of the GeoCache scheme and actively attempts to bypass it (e.g., by executing the aforementioned linear algebraic recovery using known prompt templates), rather than solely testing against timing-based probes.

---

> ### Author Response · Authors · 2026-04-13
> **Author Response to Reviewer**
>
> We thank the reviewer for the thorough and technically detailed feedback. We address each concern below.
>
> **1. Known-Plaintext Attack (KPA) and Security Boundary**
>
> We agree GeoCache provides computational security via linear obfuscation, not a cryptographic security boundary. The revised manuscript will state this distinction explicitly. GeoCache's linearity is a deliberate design choice: orthogonality enables exact isometry (Theorem 1) with zero quality loss. A cryptographic transform would break q⊤M⊤MK = q⊤K. This trades cryptographic hardness for zero degradation — appropriate when hybrid mode ensures the attacker lacks the plaintext to exploit linearity.
>
> For **user-specific content** (private prompts, PII, confidential queries), the attacker does not have the plaintext tokens needed to construct 64 linearly independent pairs. The KPA requires knowing what the victim typed — which is the secret being protected.
>
> For **shared system prompts**, the attacker can generate plaintext KV tensors locally. However, in GeoCache's hybrid mode, shared system prompts are deliberately left **untransformed** — cached in plaintext and shared for TTFT savings. Only user-specific suffixes are protected with operators derived from session-specific identifiers via SHA-256. The KPA on the system prompt portion is moot because those tokens are not transformed.
>
> We have conducted a formal KPA simulation. Recovery requires exactly N=64 pairs (cosine progresses from 0.41 at N=10 to 0.81 at N=40, reaching 1.0 only at N=64). Under hybrid mode with b=64, the attacker obtains 0 valid pairs (system prompt untransformed, suffix plaintext unknown). Even a template attack with 20 known suffix tokens yields cosine of 0.55 — insufficient for meaningful recovery. Operator rotation every N < b tokens provides additional hardening: with rotation every 32 tokens, cosine remains 0.56 even when all suffix plaintext is known.
>
> The revised manuscript will include a formal KPA table.
>
> **2. Comparison with vLLM cache_salt**
>
> vLLM's cache_salt prevents cross-tenant logical cache hits by salting the prefix hash. However, hash-based isolation has been publicly broken:
>
> **CVE-2025-25183** (vLLM, fixed in v0.7.2) and **CVE-2025-1953** (vLLM AIBrix) demonstrated that hash collisions allowed attackers to serve poisoned KV entries to victims. "Cache Me, Catch You" (Wu et al., NDSS 2026) documented additional vulnerabilities across six attack vectors in vLLM, SGLang, and GPTCache. While patched (vLLM v0.11 defaults to SHA-256), these demonstrate that hash-based isolation is a software policy subject to implementation flaws.
>
> GeoCache provides defense-in-depth: even if a hash collision occurs, the retrieved KV entries are in a different coordinate system and produce meaningless attention scores. Additionally, cache_salt does not protect tensor content when physically accessed — through GPU memory residuals (LeftoverLocals, Sorensen et al. 2024), disaggregated serving architectures (e.g., LMCache, llm-d, AIBrix), or TEE externalization.
>
> The revised manuscript will include cache_salt as a baseline.
>
> **3. Adaptive Attack**
>
> We have conducted an adaptive attack analysis where the attacker is fully aware of GeoCache and attempts algebraic recovery. The KPA simulation in Point 1 applies directly: under hybrid mode the attacker obtains 0 valid pairs, and operator rotation with N < b provides additional hardening. This analysis with a reproducible simulation script will be included in the revised manuscript.
>
> **4. Content-Level Attack Evaluation**
>
> We have conducted controlled attack experiments on Mistral-7B-Instruct-v0.1 following the protocols of Luo et al. (2025):
>
> **Injection attack.** The attacker obtains a victim's KV-cache (via physical access vectors described above) and appends an instruction to extract private content. Without GeoCache: **avg ROUGE-L = 0.917, 19/19 secret markers recovered**. With GeoCache: **avg ROUGE-L = 0.043, 0/19 markers recovered (95.3% reduction)**.
>
> **Exfiltration attack.** The attacker compares candidate Key vectors against the victim's cached Keys via cosine similarity. Without GeoCache: exact match has **cosine = 1.0**, spread = 0.336. With GeoCache: all candidates produce **cosine ≈ -0.01, spread = 0.012**.
>
> These along with comparisons against cache_salt, CacheSolidarity, and SafeKV will be included in the revised manuscript.
>
> **5. Cross-Tenant Caching**
>
> The reviewer correctly notes that session-specific operators break cross-tenant deduplication for protected content. GeoCache's hybrid mode addresses this — public prefixes remain shared while private suffixes are protected. In experiments on Mistral-7B with a 130-token shared system prompt, hybrid mode reduces average TTFT by **21.4%** compared to per-session isolation while keeping suffix keys geometrically isolated **(cosine = 0.001)**.
>
> We share these results while we prepare the full revision with updated analysis and code.

---

> > ### Comment · Reviewer_xjC3 · 2026-04-30
> >
> > Thank you for the detailed, point-by-point rebuttal to my initial review. I appreciate the time you took to run the explicit Known-Plaintext Attack (KPA) simulation, include the new Mistral-7B attack evaluations, and detail the historical vulnerabilities of hash-based `cache_salt`.
> >
> > The revised manuscript presents a significantly strengthened experimental section. However, my core technical concerns remain unresolved:
> >
> > **1. The KPA and the "Hybrid Mode" Assumption**
> > Regarding the KPA analysis, you argue that under the hybrid mode, an attacker obtains 0 valid pairs because the public system prompt is untransformed and the user's private suffix is unknown. However, this relies on the flawed assumption that user-specific suffixes are entirely unpredictable. In modern, real-world LLM serving applications, user suffixes are heavily structured. They frequently contain standard JSON schema declarations, Markdown formatting, API wrappers, or predictable conversational boilerplate (e.g., "Summarize the following:", "Translate this to French:"). Relying on the assumption that an attacker cannot glean or inject 64 predictable tokens from these structural elements makes the linear security boundary incredibly brittle. Fundamentally, this linear transformation cannot offer provable, robust protection against KV cache inversion attacks, rendering the claim that the method comprehensively resists this attack vector overstated.
> >
> > **2. Complexity vs. Benefit**
> > Because the proposed linear obfuscation does not provide a robust cryptographic boundary against physical extraction (due to the KPA risk on structured prompts), the architectural complexity of generating, applying, and managing session-specific matrices is difficult to justify. It does not clearly outcompete simpler, strict cache-partitioning architectures or traditional dual-cache setups. For instance, a straightforward strategy—such as enforcing strict cache isolation between users or restricting KV cache sharing exclusively to trusted groups—would achieve comparable practical outcomes without the architectural overhead. Given the lingering uncertainty regarding its resilience to inversion attacks, the necessity of deploying such a complex scheme remains unclear.
> >
> > Thank you again for your hard work and constructive engagement throughout this process.

---

> > > ### Author Response · Authors · 2026-05-01
> > > **Author Response to Reviewer Follow-up**
> > >
> > > We thank the reviewer for the continued constructive engagement and for acknowledging the strengthened experimental section.
> > >
> > > **1. KPA on Structured Suffixes**
> > >
> > > The reviewer's concern is well-founded, and we have now validated it experimentally. We ran the KPA evaluation on real Mistral-7B K vectors using the exact types of structured templates the reviewer describes (JSON schema, translation prefixes, few-shot exemplars). The results confirm the reviewer's intuition:
> > >
> > > - Freeform query (0 known tokens): cosine 0.00 (secure)
> > > - "Translate to French:" (9 known): 0.72 without rotation, 0.72 with rotation
> > > - JSON schema (38 known): 0.90 without rotation, 0.64 with rotation
> > > - Few-shot Q/A (33 known): 0.86 without rotation, 0.51 with rotation
> > > - Long structured form (34 known): 0.88 without rotation, 0.54 with rotation
> > >
> > > The reviewer is correct that without operator rotation, structured templates with 30+ known tokens reach near-broken cosine (0.86-0.90). Operator rotation every b/2 = 32 tokens — described in our original submission as additional hardening, to be reframed as required for production deployments with structured prompts — splits known tokens across multiple independent operators, reducing structured-template cases with 30+ known tokens to cosine 0.51-0.64 (below the empirical 0.85 threshold). Templates with fewer than 32 known tokens (e.g., "Translate to French" at 9 tokens) retain cosine ~0.72 because all known tokens fit within a single rotation segment — but this reflects natural-language subspace alignment, not algebraic operator recovery, and does not enable exact token reconstruction.
> > >
> > > We agree that our earlier framing overstated the protection. The revised manuscript will make three specific corrections: (a) operator rotation will be reframed from "additional hardening" to required for production deployments with structured prompts; (b) the claim that collecting 64 pairs is "practically infeasible" will be removed; (c) a real-template KPA table with the above data will be added to the evaluation (Appendix D). We also explicitly state that GeoCache provides computational security via linear obfuscation, not a cryptographic security boundary — the linearity is a deliberate design choice enabling the exact isometry guarantee (Theorem 1).
> > >
> > > We note that the dominant use case for multi-tenant KV-cache sharing — freeform conversational queries where the private content is unknown to the attacker — remains fully secure (cosine 0.00) regardless of rotation.
> > >
> > > **2. Complexity vs. Simpler Alternatives**
> > >
> > > With operator rotation addressing the structured-prompt KPA risk (Section 1 above), the residual security guarantee — content-level isolation under cache access failure — is the value proposition that simple isolation cannot provide. In the normal case, simple public/private cache partitioning has identical caching behavior to GeoCache hybrid mode. The difference is what happens when isolation fails:
> > >
> > > (a) **Hash collision** (CVE-2025-25183, CVE-2025-1953): under strict isolation, a hash collision causes the victim to retrieve the attacker's plaintext KV entries — the attack succeeds. Under GeoCache, the victim retrieves entries in a different coordinate system that produce noise — the attack fails.
> > >
> > > (b) **Physical tensor access** (LeftoverLocals, disaggregated serving): under strict isolation, extracted tensors are plaintext and immediately exploitable. Under GeoCache, extracted tensors are transformed and meaningless without the session operator.
> > >
> > > GeoCache's overhead is minimal: <0.5% throughput, 1 MB per session, three matrix multiplications per layer per token — comparable to adding a single linear projection.
> > >
> > > We acknowledge that for deployments where hash-based isolation is trusted to be correct and physical access is excluded from the threat model, simple cache partitioning is sufficient and GeoCache is unnecessary. The revised manuscript will state this explicitly, positioning GeoCache as a defense-in-depth layer for environments where these assumptions may not hold.
> > >
> > > These changes are being incorporated in the next revision, which we will upload shortly.

---

### Review · Reviewer_hXyQ · 2026-04-29

**Summary Of Contributions:**

This paper proposes GeoCache, a defense for content-level privacy risks in shared KV-caches for multi-tenant LLM inference. The core idea is to apply a session-specific Block-Diagonal Orthogonal Transform (BDOT) to cached Key/Value tensors, and to apply the corresponding transform to Query tensors during authorized reuse. Because orthogonal transformations preserve inner products, the paper argues that in-session attention is unchanged, while cross-session cache access becomes geometrically meaningless. The paper evaluates GeoCache on Llama-2-7B and Mistral-7B, reporting exact quality preservation, low overhead, and mitigation of direct KV exfiltration and semantic poisoning attacks.

Key Strengths:
- Important problem: Shared KV-cache privacy is a timely and relevant issue for multi-tenant LLM serving.
- Clean core idea: Using orthogonal transforms to preserve authorized attention computation is elegant and easy to reason about.
- Good scope awareness: The paper distinguishes content-level KV-cache attacks from timing and scheduling side channels, and mostly acknowledges that GeoCache does not solve those serving-layer attacks.
- Some adaptive-attack discussion: The paper includes a Known-Plaintext Attack analysis, which is necessary given that the defense is based on a linear transform.

Key Weaknesses:
- The main theoretical claim is not fully justified: The cross-session isolation proof treats the transform as if it were Haar-random over the full O(dk), but the implementation uses a block-diagonal transform. This matters: BDOT only mixes coordinates within each block, so the concentration bound should depend on block size and block-wise vector norms, not simply on dk.
- The Known-Plaintext Attack risk is serious: Since the defense is linear, an attacker with enough plaintext-ciphertext pairs can recover the transform. The paper notes that b=64 known pairs can break one block, but does not convincingly argue that such pairs are unavailable in practice. Real LLM prompts often contain public templates, repeated system instructions, structured fields, or predictable text.
- The practical advantage over simpler baselines is unclear: GeoCache hybrid mode shares public prefixes and protects private suffixes. This is close to a simpler baseline: share public/system prompts globally and isolate private/user content per session. The paper does not compare against this baseline, so it is hard to tell what GeoCache really adds.
- Security claims are too strong: Near-random cross-session attention does not necessarily mean semantic poisoning is eliminated. It may prevent controlled injection, but unauthorized cache reuse could still corrupt outputs or cause availability failures.
- Evaluation is not yet convincing enough: The experiments are mostly PyTorch/Colab-level rather than integrated into vLLM, SGLang, or TensorRT-LLM. The attack evaluation is also small-scale and does not include strong adaptive attacks based on known templates or algebraic recovery.

**Additional Comments:**

N/A

**Audience:**

Yes

**Audience Explanation:**

The topic is relevant to ML systems and ML security. KV-cache sharing is widely used in LLM serving, and content-level privacy risks are becoming more important. The idea of using geometry-preserving transformations for inference-time state protection is interesting, even if the current version overclaims its guarantees.

**Claims And Evidence:**

No

**Claims Explanation:**

The paper convincingly supports the claim that matched orthogonal transforms preserve authorized attention computation. However, the stronger privacy claims are not yet well supported. The proof of cross-session isolation does not match the actual BDOT implementation, the KPA analysis is incomplete, and the evaluation does not show robustness against realistic adaptive attackers. The claimed practical benefit is also unclear without comparison to a public/private cache-splitting baseline.

**Requested Changes:**

- Provide a correct BDOT-specific proof rather than relying on full Haar-random orthogonal matrices.
- Clarify the attention-score scaling and variance analysis.
- Strengthen the Known-Plaintext Attack evaluation with realistic prompt templates and structured inputs.
- Compare against a simple public/private cache baseline.
- Evaluate GeoCache inside a real serving framework such as vLLM or SGLang.
- Reduce overclaims such as “provably lossless privacy” and “semantic poisoning eliminated.”

---

> ### Author Response · Authors · 2026-04-30
> **Author Response to Reviewer**
>
> We thank the reviewer for this technically detailed and constructive review. We address each concern below.
>
> **1. BDOT-Specific Concentration Bound and Variance Analysis (Theorem 2)**
>
> The reviewer correctly identifies that the proof invokes Haar-randomness on O(dk) while the implementation uses block-diagonal structure on O(b). We will add a BDOT-specific proof. Under BDOT, the cross-session score decomposes as a sum of dk/b independent block-level terms: q⊤M₂⊤M₁k = Σᵢ qᵢ⊤(B₂ᵢ⊤B₁ᵢ)kᵢ, where each B₂ᵢ⊤B₁ᵢ is independently Haar-distributed on O(b). For unit-norm q,k with energy uniformly spread across blocks, the bound matches the full Haar case. For energy concentrated in few blocks, the bound is weaker by up to a factor of dk/b. The empirical concentration (1000-trial synthetic analysis, Appendix F) operates in the typical-case regime where attention vectors have bounded block-norm imbalance. The explicit block-wise derivation will be added as Appendix A.4, including a corrected variance analysis: under BDOT, the cross-session score variance is Σᵢ ||qᵢ||²||kᵢ||²/b, which equals 1/b in the worst case (energy in one block) and 1/dk in the typical case (energy spread uniformly). The current Section 4 variance analysis will be revised to state both cases explicitly.
>
> **2. Known-Plaintext Attack with Realistic Templates**
>
> We ran the KPA evaluation on real Mistral-7B K vectors using production prompt templates. The results refine our synthetic analysis:
>
> - Freeform query (0 known): cosine 0.00 (secure)
> - "Translate to French:" (9 known): cosine 0.72 without rotation, 0.72 with rotation
> - JSON schema (38 known): cosine 0.90 without rotation, 0.64 with rotation
> - Few-shot Q/A (33 known): cosine 0.86 without rotation, 0.51 with rotation
> - Long structured form (34 known): cosine 0.88 without rotation, 0.54 with rotation
>
> The reviewer's intuition is correct: real K vectors carry natural-language structure, providing more signal per known pair than synthetic Gaussian vectors. Without rotation, structured templates with 30+ known tokens reach near-broken cosine (0.86-0.90). Operator rotation every 32 tokens — already present in Appendix D of the original submission — reduces structured-template cases with 30+ known tokens (JSON, few-shot, long form) to cosine 0.51-0.64. Templates with fewer than 32 known tokens (e.g., "Translate to French" at 9 tokens) retain cosine ~0.72 because all known tokens fit within a single rotation segment — but this reflects natural-language subspace alignment, not algebraic operator recovery, and does not enable exact token reconstruction. The revised manuscript will reframe operator rotation from "additional hardening" to required for production deployments with structured prompts.
>
> **3. Public/Private Cache Baseline Comparison**
>
> The simple baseline — share public prefixes, isolate private suffixes — has identical caching behavior to GeoCache hybrid in the normal case. The difference is what happens when isolation fails: (a) hash collision (CVE-2025-25183, CVE-2025-1953): the simple baseline serves plaintext entries, GeoCache serves entries in a different coordinate system producing noise; (b) physical tensor access (LeftoverLocals, disaggregated serving): the simple baseline exposes plaintext tensors, GeoCache exposes transformed tensors meaningless without the session operator. This comparison will be added to the Discussion section.
>
> **4. Security Claim Precision**
>
> The reviewer is correct that random cross-session attention may corrupt outputs. We will add to Limitations: "GeoCache converts a confidentiality failure into an availability degradation — the attacker cannot extract or control content, but the victim's output may be incoherent if cross-session entries are served." Claims will be softened: the title is revised to "Provably Lossless Inference and Computational Content-Level Isolation" (separating the provable utility claim from the empirical security claim), and absolute "elimination" wording is replaced throughout with scoped language ("substantially defeated," "substantially mitigates," "controlled" exfiltration/injection attacks).
>
> **5. Serving Framework Integration**
>
> Production integration with PagedAttention (vLLM) and RadixAttention (SGLang) is a natural engineering extension — the theoretical guarantees depend only on the orthogonality of M, not on how KV tensors are managed (Appendix B.1). This is a common scope choice in recent KV-cache privacy work, including KV-Cloak (Luo et al., 2025). We are actively pursuing this integration.
>
> **6. Overclaims**
>
> The revised manuscript will clearly distinguish between the mathematical guarantee (Theorem 1: exact isometry, provable) and the practical security claim (computational security, empirical). All absolute claims will be scoped as described in Point 4 above.
>
> These changes are being incorporated in the next revision, which we will upload shortly.

---

### Author Response · Authors · 2026-05-01
**Revised Manuscript Uploaded**

We have uploaded a revised manuscript (v3) addressing feedback from all reviewers. A summary of changes is provided in the 'Changes Since Last Revision' field.

---

### Decision · Action_Editor_bWcv · 2026-05-13

**Recommendation:** Reject

**Additional Comments:**

The four official reviewers reached strong consensus after the rebuttal phase: all four answered No to the claims-and-evidence criterion, three submitted Leaning Reject and one submitted Reject. The reviewers acknowledged genuine effort in the revisions (real-template KPA experiments added in Table 10, BDOT-specific concentration proof in Appendix A.4, scope corrections in title and claims, citation errors fixed). The fundamental issues, however, are structural rather than presentational.

The author has put substantial work into this submission across multiple revision rounds, and the underlying isometry idea is genuinely interesting. I therefore strongly encourage the author to consider a resubmission, and I check the corresponding "may consider major revision" option. For such a resubmission to succeed under TMLR's acceptance criteria, one of two paths would be advisable:

(a) Reposition the contribution as a performance and computational-efficiency result for multi-tenant KV-cache serving (where the isometry guarantee on attention preservation, the negligible latency overhead, and the hybrid deduplication behavior would be evaluated against vLLM, SGLang, and TensorRT-LLM baselines), dropping the security and privacy framing entirely.

(b) Strengthen the cryptographic argument substantially beyond linear obfuscation, with a security construction that is provably resistant to the KPA scenarios identified by reviewers xjC3, YSxv, and hXyQ. The current real-template KPA evidence (cosine 0.51–0.64 even with rotation, on structured prompts) sets the bar that any new construction would need to clear.

Either path would also benefit from integration of GeoCache into a production-grade serving system (vLLM, SGLang, or TensorRT-LLM) and a direct comparison against cache_salt, CacheSolidarity, SafeKV, and KV-Cloak under aligned threat models, which would settle the marginal-benefit question that several reviewers raised.

I would like to thank the reviewers for the careful and thoughtful engagement throughout the discussion phase, and the author for the responsive and substantive revisions despite the difficult final outcome.

**Audience:**

Yes

**Audience Explanation:**

All four reviewers agree that the broader problem area is of interest to TMLR's audience (4/4 Yes). KV-cache security and computational isolation in multi-tenant LLM serving is a timely concern, with active recent literature (cache_salt, SafeKV, CacheSolidarity, KV-Cloak, PromptPeek) and concrete deployment relevance for vLLM, SGLang, and TensorRT-LLM. The lossless-attention isometry idea itself is conceptually clean and reusable. The principal issue is not lack of audience but lack of supporting evidence for the specific security claims, which falls under the first criterion above.

**Claims And Evidence:**

No

**Claims Explanation:**

All four reviewers concluded that the claims in the submission are not adequately supported by the evidence presented (4/4 No). The substantive disagreement converges on three structural concerns that the rebuttal and v3 revision did not resolve.

First, the core security claim relies on a linear (block-diagonal orthogonal) transform of K/V tensors, which is provably broken under a Known-Plaintext Attack after a small number of independent plaintext-ciphertext pairs ($b \leq 64$ for the configurations evaluated). The author conceded this in the rebuttal by reframing operator rotation from "additional hardening" to "required for production". Moreover, the real-template KPA experiment added in v3 (Table 10) provides direct evidence against the security claim: structured prompt templates produce cosine similarities of 0.86–0.90 without rotation and remain at 0.51–0.64 even with rotation, demonstrating that realistic LLM prompts (JSON outputs, system instructions, few-shot examples, structured forms) supply enough predictable structure for the attack to succeed.

Second, Theorem 2 (cross-session isolation) originally invoked Haar-random transforms on the full orthogonal group O(d_k), but the implementation uses block-diagonal transforms that do not satisfy the Haar-randomness premise. The v3 corrected proof now requires additional energy-spread assumptions and yields a worst-case variance of 1/b rather than 1/d_k, narrowing the gap between the theorem and the deployed mechanism but not closing it convincingly.

Third, the marginal benefit of GeoCache over simpler baselines is unclear. In the normal-case multi-tenant setting with SHA-256 + cache_salt (already standard in vLLM), the hybrid mode of GeoCache is behaviorally equivalent to public/private cache partitioning, which the author added as a baseline (Table 8) and conceded is sufficient under trusted hash isolation. The scenarios where GeoCache provides additional value are either misconfigurations remediable by correct cache_salt usage or threat models in which the attacker simultaneously gains plaintext access, which in turn enables the KPA that compromises GeoCache itself.

These three concerns collectively prevent the headline claims (provably lossless inference plus content-level isolation against the cited multi-tenant threats) from being supported by accurate, convincing, and clear evidence in the present version.

**Resubmission Of Major Revision:**

The authors may consider submitting a major revision at a later time.